# Autocrine interferon poisoning mediates ADAR1-dependent synthetic lethality in *BRCA1/2*-mutant cancers

ADAR1 is an RNA editing enzyme which prevents autoimmunity by blocking interferon responses triggered by cytosolic RNA sensors, and is a potential target in immuno-oncology. However, predictive biomarkers for ADAR1 inhibition are lacking. Using multiple in vitro and in vivo systems, we show that *BRCA1/2* and *ADAR1* are synthetically lethal, and that ADAR1 activity is upregulated in *BRCA1/2*-mutant cancers. ADAR1 depletion in *BRCA1*-mutant cells causes an increase in R-loops and consequently, an upregulation of cytosolic nucleic acid sensing pattern recognition receptors (PRR), events which are associated with a tumor cell-autonomous type I interferon and integrated stress response. This ultimately causes autocrine interferon poisoning. Consistent with a key role of R-loops in this process, exogenous RNase H1 expression reverses the synthetic lethality. Pharmacological suppression of cell-autonomous interferon responses or transcriptional silencing of cytosolic nucleic acid sensing PRR are also sufficient to abrogate *ADAR1* dependency in *BRCA1*-mutant cells, in line with autocrine interferon poisoning playing a central part in this synthetic lethality. Our findings provide a preclinical rationale for assessing ADAR1-targeting agents in *BRCA1/2*-mutant cancers, and introduces a conceptually novel approach to synthetic lethal treatments, which exploits tumor cell-intrinsic cytosolic immunity as a targetable vulnerability of cancer cells.

Adenosine Deaminase Acting on RNA 1 (ADAR1) catalyzes the conversion of adenosines to inosines in double-stranded RNA (dsRNA) substrates – a post-transcriptional process referred to as "A-to-I RNA editing"[1]. ADAR1 is part of a family of three paralogs that share a common structural backbone, consisting of two or three N-terminal dsRNA-binding domains and one C-terminal catalytic deaminase domain. ADAR1 uniquely harbors two Z-DNA binding domains, conferring the ability to recognize the left-handed helical variant of DNA (Z-DNA) in a sequence-independent manner. ADAR1 exists as two isoforms: (i) a full-length, interferon-stimulated ADAR1p150 isoform predominantly localized in the cytoplasm (which potentially shuttles to and from the nucleus); and (ii) an N-terminally truncated ADAR1p110 isoform exclusively localized in the nucleus[1].

The canonical function of ADAR1 is editing of dsRNA species derived primarily from non-coding genomic regions (notably introns and 3′-UTRs harboring Alu elements) and, less frequently, from protein-coding regions of mRNAs[2]. By carrying out this activity, cytoplasmic ADAR1p150 prevents autoimmune responses to endogenous transcripts by limiting the extent of dsRNA sensing mediated by pattern recognition receptors (PRR) such as RIG1, MDA5, LGP2, PKR, or ZBP1[3]. ADAR1, particularly ADAR1p110 isoform, has also been found to maintain genome stability by promoting the resolution of R-loop at telomeres, stalled replication forks and DNA double-strand breaks (DSBs)[4-6]. These ostensibly distinct functions of ADAR1 link the DNA damage response (DDR) and anti-tumor immunity[7,8].

✉ e-mail: roman.chabanon@gustaveroussy.fr; chris.lord@icr.ac.uk; sophie.postel-vinay@gustaveroussy.fr

ADAR1 is an attractive therapeutic target in immuno-oncology, on the basis that inhibiting ADAR1 could increase tumor immunogenicity by suppressing RNA editing[9,10]. Previous studies have proposed that triple-negative breast cancer (TNBC) cells with a heightened interferon-stimulated gene signature are sensitive to *ADAR1* gene silencing[11–13] as are tumor cells with elevated levels of dsRNA species[14,15]. However, as yet, no clinically-actionable predictive biomarkers for ADAR1 inhibition sensitivity have been discovered.

In this work, we uncover a robust and penetrant synthetic lethality (SL)[16] between *BRCA1/2* and *ADAR1*. In contrast to other *BRCA1/2* SL effects – such as that mediated by PARP inhibitors which relies on DNA damage, cGAS-STING pathway activation and T-cell-mediated adaptive immune response[17,18], we show that the *BRCA1/2–ADAR1* SL is a tumor cell-autonomous effect not only caused by R-loops, but importantly, by a resultant PRR activation leading to autocrine interferon poisoning.

## Results

### A *BRCA1/2–ADAR1* synthetic lethality exists both in vitro and in vivo

Our prior work, and that of others, has shown that *BRCA1/2*-mutant (*BRCA*m) tumor cells display elevated cGAS-STING signaling that can be exacerbated by exposure to PARP inhibitors (PARPi)[17–21]. Although this relationship has been established using both in vitro and in vivo models of cancer, there is no evidence of *BRCA*m tumor cells being reliant upon cGAS-STING signaling for survival, and inactivation of cGAS-STING signaling only alters the response to PARPi in vivo in presence of a functional immune system and not in vitro. This indicates that the contribution of cGAS-STING signaling to the *BRCA1/2*–PARPi SL is not tumor cell-autonomous (i.e., it relies on non-tumor cell lineages) and is best explained by cGAS-STING signaling activating a T-cell-mediated immune response[17,18]. We hypothesized that, unlike cGAS or STING, other nucleic acid sensing pattern recognition receptors (PRR) might be essential for *BRCA*m tumor cells. To assess this, we carried out a focused small-interfering RNA (siRNA) screen to identify synthetic lethal (SL) interactions between nucleic acid sensing PRR and *BRCA1*. In this screen, we used a *BRCA1*-isogenic system consisting of the *BRCA1*-mutant (*c.2288delT*, p.N723fsX13), SUM149 triple-negative breast cancer (TNBC) cell line (SUM149 *BRCA1*-Mut) and a previously described daughter clone, SUM149 *BRCA1*-Rev, with a secondary reversion mutation in *BRCA1* (*c.[2288delT;2293del80])* that restores *BRCA1* function[22] and PARPi resistance (Fig. 1A, and Supplementary Fig. 1A). Cells were reverse transfected with a siRNA library targeting 18 PRR-encoding genes and cell viability was estimated six days later (Fig. 1B). siRNA targeting *ADAR1*, but none of the other PRR-targeting siRNAs, elicited *BRCA1* SL (Fig. 1C; median surviving fraction of 20.4% in SUM149 *BRCA1*-Mut *vs.* 96.7% in SUM149 *BRCA1*-Rev cells; $P < 0.0001$, two-way ANOVA).

In validation experiments, we found that the transfection of *ADAR1*-targeting siRNA SMARTPool (siADAR1-P; a pool of four different siRNAs targeting *ADAR1*, which was used in the screen) as well as two individual *ADAR1* siRNAs (siADAR1-#1 and -#2, deconvoluted from the pool) silenced both ADAR1p110 and ADAR1p150 isoforms (Supplementary Fig. 1B) and selectively inhibited clonogenic survival of SUM149 *BRCA1*-Mut cells when compared to SUM149 *BRCA1*-Rev cells (Fig. 1D, E; and Supplementary Fig. 1C, D). To eliminate the possibility of an off-target effect of the siRNAs, we carried out two orthogonal experimental approaches to target *ADAR1* in SUM149 *BRCA1*-isogenics: (i) CRISPR-Cas9 mutagenesis of *ADAR1* using four different single-guide RNAs (sgRNAs); and (ii) transcriptional silencing of ADAR1 using a doxycycline-inducible short-hairpin RNA (shRNA). Both approaches depleted ADAR1 levels and led to a reduction in clonogenic survival, cell viability and cell proliferation of SUM149 *BRCA1*-Mut cells, while SUM149 *BRCA1*-Rev cells were less affected (Fig. 1F, G; and Supplementary Fig. 1E–L). The extent of these effects was proportionate to

the level of ADAR1 silencing (Supplementary Fig. 1I–L), suggesting a correlation between cell survival and residual ADAR1 expression in *BRCA1*-mutant cells.

Many SL effects have incomplete penetrance, i.e., are limited to a small number of models, fail to operate in molecularly-diverse backgrounds and/or are often restricted to specific cancer histotypes[16]. To explore whether the *BRCA1–ADAR1* SL effect was private to SUM149 cells or more penetrant, we assessed the effects of siRNA- or CRISPR-Cas9-mediated targeting of *ADAR1* in ten additional independent models (Fig. 2A): (i) four isogenic systems of *BRCA1* deficiency, including mouse (*Mus musculus*, Mm) embryonic fibroblasts (MEFs)[23], human retinal pigment epithelial cells (RPE1)[24], mouse mammary carcinoma cells (4T1)[25] or mouse ovarian carcinoma cells (ID8)[26]; and (ii) a molecularly diverse, non-isogenic panel of six human TNBC cell lines with/without endogenous *BRCA1* mutations (*BRCA1*-wildtype: MDA-MB-231, CAL51, CAL120, Hs578T; *BRCA1*-mutant: MDA-MB-436, HCC1937). The homologous recombination (HR) status of these cell lines was confirmed by assessing PARPi sensitivity (Supplementary Figs. 1–2). In all models, *BRCA1*-mutant cells were significantly more sensitive to ADAR1 silencing than *BRCA1*-wildtype cells (Fig. 2B–D; and Supplementary Fig. 1N; Supplementary Fig. 2). In assessing whether *ADAR1* SL effects extended to *BRCA2*-mutant cells, we found that ADAR1 silencing elicited SL in two different *BRCA2*-isogenic systems: human *BRCA2*-knockout colorectal carcinoma cells (DLD1)[27] and mouse *Brca2*-knockout 4T1 cells[25] (Fig. 2D; and Supplementary Fig. 2G–J, 3A-F). We also confirmed the *BRCA1/2–ADAR1* SL effect in isogenic *ADAR1*-wildtype or -knockout human embryonic kidney cells (HEK293T) subjected to BRCA1 or BRCA2 silencing[28] (Fig. 2D–F).

Considering the predominant role of ADAR1p150 isoform in mediating RNA editing[29] and averting dsRNA sensing-driven autoimmunity[30], we hypothesized that the loss of ADAR1p150 function might be sufficient to elicit SL in *BRCA*m tumor cells. Short-term survival assays in SUM149 *BRCA1*-isogenics subjected to transfection with an ADAR1p150-specific siRNA revealed that ADAR1p150 silencing phenocopied the SL elicited by pan-isoform *ADAR1* siRNAs in *BRCA1*-mutant cells (Supplementary Fig. 3G, H). Similar SL effects were also observed in an ADAR1p150-null (p110-intact) HEK293T clone[28] subjected to BRCA1 or BRCA2 silencing to the same extent as in isogenic *ADAR1*-knockout cells (Supplementary Fig. 3I–L), corroborating the existence of a *BRCA1/2–ADAR1p150* SL. To further explore the relative contribution of ADAR1 isoforms to the SL, we conducted short-term survival assays in SUM149 *BRCA1*-isogenic cells subjected to (i) ADAR1 knockdown with an siRNA targeting *ADAR1* 3'-UTR region and (ii) concomitant exogenous overexpression of ADAR1p110 or ADAR1p150. We found that overexpression of ADAR1p150, but not ADAR1p110, could reverse the *BRCA1–ADAR1* SL effects (Supplementary Fig. 3M, N), supporting a key role for ADAR1p150 in this SL. To next investigate whether ADAR1p150 deaminase activity was involved in the SL, we transfected plasmids encoding wildtype ADAR1p150, the deaminase-defective mutant G1007R (p150-ΔCD), or the Z-DNA binding-defective mutant P193A (p150-ΔZα)[31] in SUM149 models. We observed that ADAR1p150-ΔZα, but not ADAR1p150-ΔCD, significantly reversed the SL effects (Supplementary Fig. 3O), indicating the essentiality of ADAR1p150 deaminase activity in *BRCA1*-mutant cells survival.

In mice, constitutive *Brca2* deletion is embryonically lethal[32], whereas in zebrafish (*Danio rerio*, Dr), its orthologue *brca2*[33] can be inactivated where it causes a HR defect and PARPi sensitivity[34,35]. Moreover, the zebrafish *adar1* gene conserves the A-to-I RNA editing[36,37] and protection against aberrant interferon signaling[38] functions, similar to human *ADAR1*. This allowed us to assess the *BRCA2–ADAR1* SL in a whole organism system, using a translation-blocking morpholino (MO) approach[39] (Fig. 2G–J). In fish injected with either *brca2* or *adar1* MO alone, a large majority of offspring survived and developed normally (alive / normal morphology: *brca2* MO, 93% / 87%; *adar1* MO, 86% / 65%; control MO, 96% / 95%; Fig. 2G, I). In

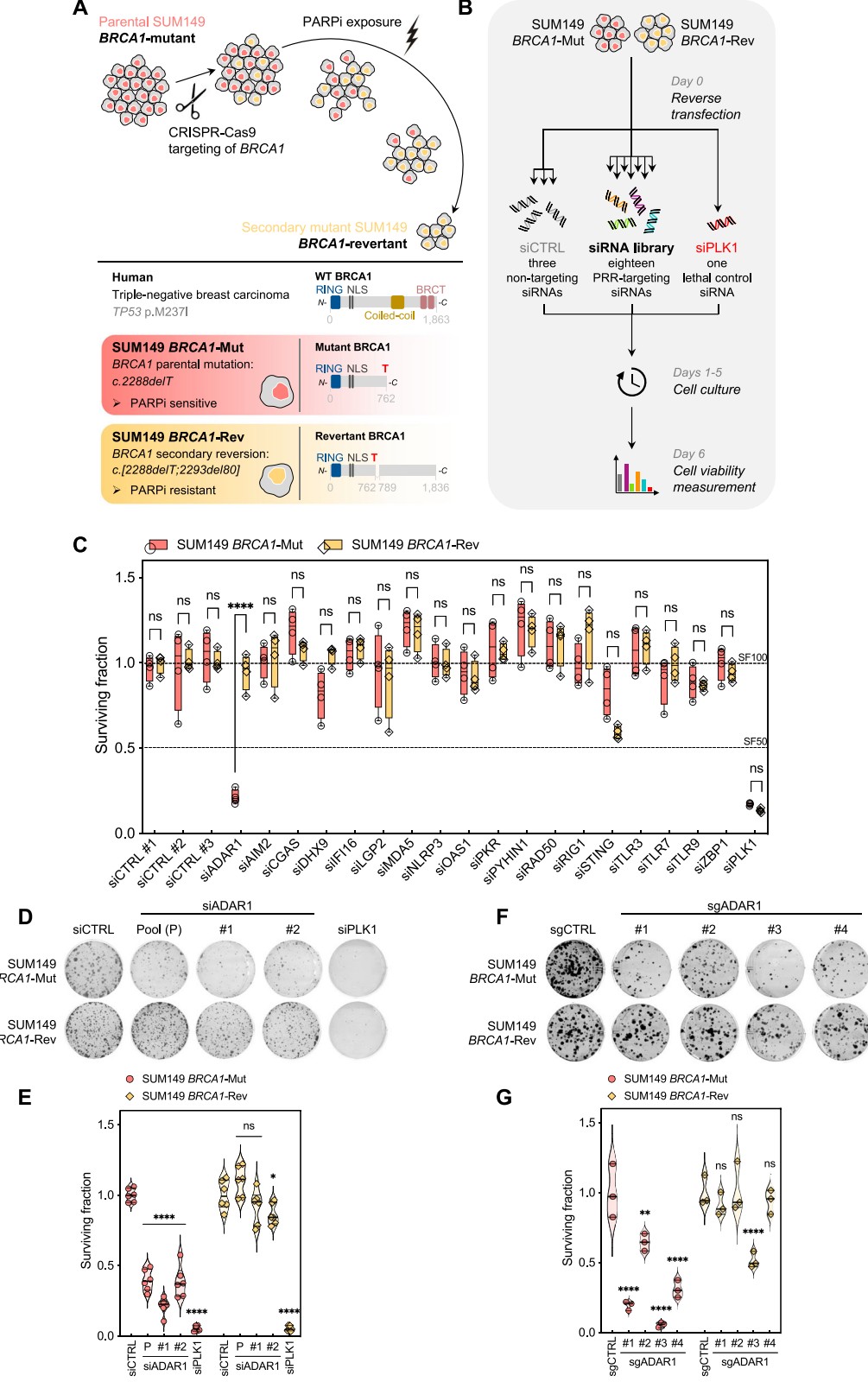

contrast, in fish injected concomitantly with *adar1* and *brca2* MOs, offspring showed a significantly higher percentage of mortality (48%), and all remaining living larvae showed different degrees of abnormal morphology (mild, 11%; severe, 41%; Fig. 2G, I) associated with apoptosis, as assessed by acridine orange staining (Fig. 2H, J).

Altogether, this data showed that the *BRCA1/2–ADAR1* SL operates in vitro and in vivo across a variety of model systems (Fig. 2D), and that

the concomitant loss of *adar1* and *brca2* in zebrafish is embryonically lethal.

## ADAR1-mediated RNA editing is upregulated in *BRCA1/2*-mutant cancers

Given the increased reliance of *BRCA*m cells upon *ADAR1* function, we next sought evidence for a dysregulation of ADAR1 activity in *BRCA*m

**Fig. 1 | A functional screen identifies *BRCA1*–*ADAR1* synthetic lethality in triple-negative breast cancer. A** Schematic showing experimental design for the generation of SUM149 *BRCA1*-isogenic cell lines. **B, C** A focused siRNA screen (**B**) identifies *BRCA1*–*ADAR1* synthetic lethality in the SUM149 *BRCA1*-isogenic model (**C**). Box-and-whiskers indicate median, lower and upper quartiles, and the min to max range; *n* = 4 biological replicates, two-way ANOVA post hoc Šidák's test. *P* value, ****< 0.0001. **D, E** Clonogenic survival of SUM149 *BRCA1*-Mut and *BRCA1*-Rev cells transfected with *ADAR1* siRNA (P, Pool; #1; #2). siCTRL, non-targeting, negative control siRNA; siPLK1, *PLK1*-targeting, positive control siRNA. Violin plots indicate median, lower and upper quartiles; N = 6 values from individual wells,

representative of *n* = 3 biologically-independent experiments, two-way ANOVA post hoc Dunnett's test. *P* values, *=0.0196, ****< 0.0001. **F, G** Clonogenic survival of SUM149 *BRCA1*-Mut and *BRCA1*-Rev cells transfected with *ADAR1* sgRNA (#1; #2; #3; #4). sgCTRL, non-targeting, negative control sgRNA. Violin plots indicate median, lower and upper quartiles; N = 3 values from individual wells, representative of *n* = 3 biologically-independent experiments, two-way ANOVA post hoc Dunnett's test. *P* values, **=0.0015, ****< 0.0001. Source data are provided as a Source Data file. Elements of panels **A** and **B** were provided by Servier Medical Art (https://smart.servier.com/) and BioRender (https://www.biorender.com/), licensed under CC BY 4.0 (https://creativecommons.org/licenses/by/4.0/).

cancers. To this aim, we conducted a series of analyses assessing ADAR1 expression and activity in patient-derived samples of TNBC and prostate cancer – two of the cancer types in which *BRCA1/2* mutations are the most prevalent. First, we built a cohort of 63 patients with TNBC (including 32 *BRCA1*-wildtype and 31 *BRCA1*-mutant tumor samples) and optimized an immunohistochemistry assay to evaluate nuclear vs. cytoplasmic ADAR1p150 expression in tumor cells. This revealed that *BRCA1*-mutant tumors displayed a higher cytoplasmic ADAR1p150 expression (Fig. 3A, B; *P* = 0.0359, Mann-Whitney U test), while no significant difference was observed regarding ADAR1p150 nuclear expression (Supplementary Fig. 4A; *P* = 0.3976, not significant). Interestingly, higher cytoplasmic ADAR1p150 expression was also associated with increased tumor-infiltrating lymphocytes, regardless of the *BRCA1* gene status (Fig. 3C; Supplementary Fig. 4B–E). As the upregulation of *BRCA1/2* synthetic lethal partners is a general feature of HR-defective cancers[40], one reasonable assumption is that this higher ADAR1 expression seen in *BRCA1*-mutant cancers reflects the dependency these cancers have upon *ADAR1* function.

Previous studies have shown that ADAR1-mediated RNA editing was elevated in breast cancer as compared with matched, normal breast tissue[41,42] and compared with other types of cancers[41], but the impact of *BRCA1/2* mutations on the magnitude of A-to-I RNA editing in cancer remains unknown. To further investigate a functional link between the *BRCA*m genotype and ADAR1 activity, we sought to measure RNA editing levels (i.e., the frequency of A-to-I editing events in RNA transcripts, used as a surrogate marker of ADAR1 activity) in *BRCA1/2*-isogenic cell lines or patient-derived samples. To do this, we first generated RNA-Seq data of SUM149 *BRCA1*-isogenic cells exposed to ADAR1 silencing by transfection of siADAR1-P or siADAR1-#1 (Supplementary Fig. 4F) and used two complementary approaches to identify A-to-I RNA editing events from raw RNA-Seq reads (Fig. 3D, E)[43,44]. As expected, silencing of ADAR1 caused a dramatic reduction of RNA editing in both *BRCA1*-mutant and -revertant cells as measured by a decreasing A-to-I RNA Editing Index (REI; Fig. 3F) and fewer A-to-I RNA Editing Sites (RES; Fig. 3G). More importantly, these analyses revealed a significantly higher A-to-I RNA editing activity of *BRCA1*-mutant cells compared with *BRCA1*-revertant cells (mean number of 143,169 vs. 62,630 RES, respectively; *P* < 0.0001, two-way ANOVA; Fig. 3F, G), despite similar ADAR1 protein levels (Supplementary Figs. 1B, D, 4F). This difference was further specific to A-to-I RNA editing, as levels of editing observed for all other types of editing remained extremely low and unchanged (mean number of RES < 1,200 for C-to-T transitions and close to null for all transversions; Supplementary Fig. 4G-P). To assess whether such correlation would also operate in patients, we took advantage of two previously described isogenic patient-derived xenograft (PDX) models of prostate cancer – termed MR-0009 and MR-0191[45], characterized by the presence of *BRCA2* germline and secondary reversion mutations conferring sensitivity or resistance to PARPi, respectively (Fig. 3H). As in SUM149 *BRCA1*-isogenics, *BRCA2*-mutant PDXs exhibited a greater REI / more RES than their isogenic *BRCA2*-revertant counterparts (Fig. 3I, J), despite similar *ADAR1* transcript levels (MR-0009, 98.7 vs. 84.5 TPM; MR-0191, 48.4 vs. 50.8 TPM for *BRCA2*-Mut vs. *BRCA2*-Rev, respectively).

## ADAR1 suppresses DNA damage in *BRCA1/2*-mutant cells by preventing R-loop accumulation

Since the vast majority of *BRCA1/2* SL effects involve DNA repair proteins (e.g., PARP1, POLQ, APEX2, FEN1, CIP2A)[46] and result from an impaired DDR[47], we hypothesized that the causes of the *BRCA1/2*–*ADAR1* SL may also reside in an inability to process certain types of DNA damage. Indeed, we found that ADAR1 silencing in *BRCA*m cells of three different isogenic models (SUM149, MEF and 4T1) elicited nuclear γ-H2AX foci (Fig. 4A–D; and Supplementary Fig. 5A–H) and in SUM149 *BRCA1*-Rev cells elicited RAD51 foci (Supplementary Fig. 5C–E), suggesting that the loss of ADAR1 caused DNA damage mainly in *BRCA*m cells, and invoked RAD51-mediated HR repair in *BRCA1*-revertant cells. ADAR1 silencing also caused micronucleation in *BRCA*m but not in *BRCA1*-revertant or *BRCA1/2*-wildtype cells (Fig. 4E-H; Supplementary Fig. 5I), suggesting a selective induction of chromosomal instability in *BRCA*m cells. We further noted that ADAR1 silencing in *BRCA1*-mutant cells caused a significant accumulation of RPA foci in S-phase population (Fig. 4I, J; and Supplementary Fig. 5J), increased CHK1 phosphorylation and PARP1 cleavage as assessed by western blot (Fig. 4K, L; and Supplementary Fig. 5K, L), while *BRCA1*-revertant cells showed little or no change in these marks, suggesting increased replication stress and apoptosis in *BRCA1*-mutant cells subjected to *ADAR1* knockdown.

ADAR1 was recently found to facilitate R-loop resolution via A-to-I editing of RNA:DNA hybrids[4–6,48]. To explore whether an increase in R-loops could explain the *BRCA1/2*–*ADAR1* SL, we overexpressed RNase H1 (RH1), one of the main endonucleases responsible for R-loop degradation. RH1 overexpression in *BRCA1*-mutant cells reversed their sensitivity to ADAR1 silencing in colony-formation assay (Fig. 5A, B) and to a lesser extent in short-term survival assays (Supplementary Fig. 6A–E). Consistent with recent reports[4,6,49], immunofluorescence assays revealed that upon ADAR1 silencing, *BRCA1*-mutant cells demonstrated a selective increase in the formation of RNase H-sensitive S9.6 foci, and of aberrantly-shaped or ectopic nucleoli (Fig. 5C, D; Supplementary Fig. 6F-J), suggesting an accumulation of R-loops and nucleoli disruption, as previously described in other contexts[49]. Interestingly, RH1 overexpression also partially reversed the γ-H2AX, RPA foci and micronuclei phenotypes observed in *BRCA1*-mutant cells transfected with *ADAR1* siRNA, and likewise, the RAD51 foci phenotype in *BRCA1*-revertant cells (Fig. 5E; and Supplementary Fig. 7), implying a contribution of R-loops to the DDR elicited by *ADAR1* knockdown.

Altogether, these results indicated that ADAR1 protects cells from R-loop-mediated DNA damage and genomic instability, and that such function is critical for the survival of *BRCA*m cells.

## ADAR1 hedges against pattern recognition receptors-dependent, cell-autonomous interferon poisoning in *BRCA1/2*-mutant cells

Although R-loops appeared to play a key role in the *BRCA1/2*–*ADAR1* SL, we asked whether other consequences of ADAR1 inhibition may also be involved. As ADAR1 suppresses dsRNA sensing innate immune responses through RNA editing[3], and tumor cell lines with an elevated interferon-stimulated gene (ISG) signature are sensitive to

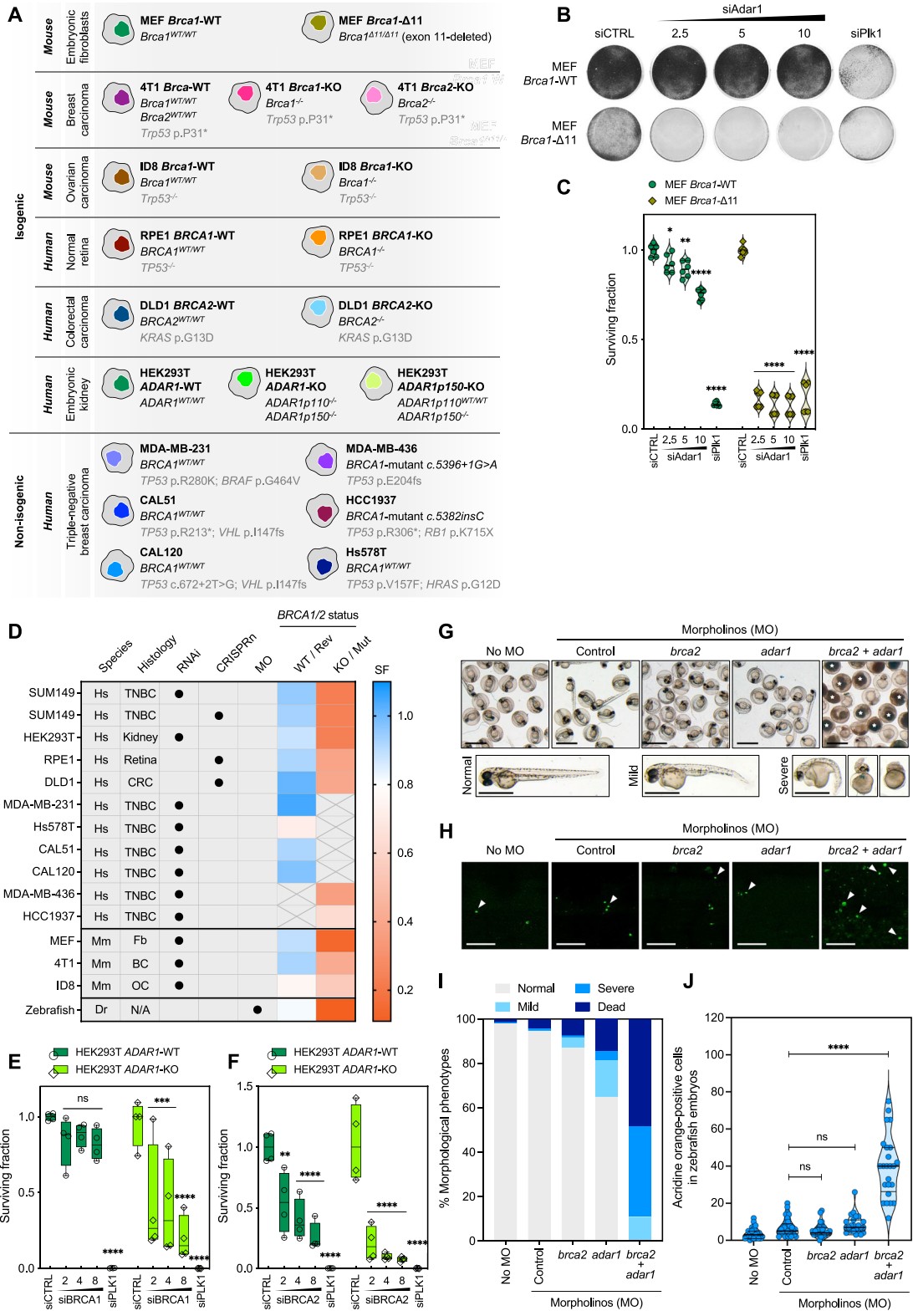

ADAR1 silencing[11–13], we sought to test whether cytosolic nucleic acid sensing might also contribute to the *BRCA1/2–ADAR1* SL effects. First, we measured in three different *BRCA1/2*-isogenic models (SUM149, MEF and 4T1) the expression and/or phosphorylation levels of proteins involved in ADAR1-dependent immunity, including (i) key dsRNA-sensing PRR (RIG1, MDA5, LGP2, PKR); (ii) markers of the integrated stress response (ISR; p-PKR, p-eIF2α); and (iii) markers of the type I interferon response (p-IRF3, p-STAT1). Western blots revealed an

increased expression of dsRNA-sensing PRR and phosphorylation of PKR, eIF2α, IRF3 and STAT1 in *BRCA*m cells but not in their isogenic *BRCA1*-revertant or *BRCA1/2*-wildtype counterparts, accompanied by selectively increased transcriptional levels of *IFNB1*, *CCL5*, and other ISGs in *BRCA*m cells of all three models, as assessed by RT-qPCR (Fig. 6A–F; and Supplementary Fig. 8A–E). The latter phenotype was also seen in *BRCA1*-mutant MDA-MB-436 cells but not *BRCA1*-wildtype MDA-MB-231 cells (Supplementary Fig. 8F, G). We noted increased IRF3

**Fig. 2 | ADAR1 silencing is synthetically lethal with *BRCA1/2* mutations in multiple model systems in vitro and in vivo. A** Schematic describing the isogenic and non-isogenic cell line models used throughout the study. **B, C** Clonogenic survival of MEF *Brca1*-wildtype (WT) and *Brca1*-mutant (Δ11) cells transfected with a concentration range (nM) of *Adar1* siRNA. Violin plots indicate median, lower and upper quartiles; N = 6 values from individual wells, representative of *n* = 3 biologically-independent experiments, two-way ANOVA post hoc Dunnett's test. *P* values, *=0.0402, **=0.0017, ****< 0.0001. **D** Heatmap showing surviving fractions elicited by ADAR1 suppression in models evaluated in Figs. 1, 2 and Supplementary Fig. 1–3 (blue, SF > 0.8; red, SF < 0.8). **E, F** Cell survival of HEK293T *ADAR1*-wildtype (WT) and *ADAR1*-knockout (KO) cells transfected with a concentration range (nM) of *BRCA1* (**E**) or *BRCA2* (**F**) siRNA. Box-and-whiskers indicate median, lower and upper quartiles, and the min to max range; N = 4 values from individual wells, representative of *n* = 3 biologically-independent experiments, two-way ANOVA post hoc Dunnett's test. *P* values, **[*ADAR1*-WT, siBRCA2 2 nM]=0.001 (**F**), ***[*ADAR1*-KO, siBRCA2 2 nM]=0.0009 (**E**), ***[*ADAR1*-KO, siBRCA1, 4 nM]=0.0005 (**E**), ****<0.0001. **G–J** Representative images and quantifications of morphological phenotypes (**G, I**) and acridine orange staining (**H, J**) in zebrafish embryos subjected to morpholino (MO)-mediated knockdown of *brca2* and/or *adar1*. Dead embryos are indicated with an asterisk (**G**). White arrows indicate acridine orange-positive cells on images taken within a defined region along the anterior-posterior axis (**H**); scale bar, 500 μm. Percentages of morphological phenotypes (**I**) were calculated based on N = 149 [no MO], N = 95 [control MO], N = 109 [*brca2* MO], N = 97 [*adar1* MO] and N = 174 [*brca2/adar1* MOs] values from individual embryos, representative of *n* = 3 biologically-independent clutches. Violin plots indicate median, lower and upper quartiles; N = 31 [no MO], N = 33 [control MO], N = 25 [*brca2* MO], N = 23 [*adar1* MO] and N = 24 [*brca2/adar1* MOs] values from individual embryos, representative of *n* = 3 biologically-independent experiments, Kruskal-Wallis test post hoc Dunn's test. *P* value, ****<0.0001. siCTRL, non-targeting, negative control siRNA; siPLK1, *PLK1*-targeting, positive control siRNA. Source data are provided as a Source Data file. Elements of panel **A** were provided by Servier Medical Art (https://smart.servier.com/), licensed under CC BY 4.0 (https://creativecommons.org/licenses/by/4.0/).

phosphorylation levels of SUM149 *BRCA1*-Mut in response to ADAR1 shRNA expression, even in absence of doxycycline (Fig. 6A), suggesting potential promoter leakage of the shRNA in that cell line. Secondly, to further explore the selective activation of an interferon response in *BRCA1*-mutant cells, we compared the ISG signature of SUM149 *BRCA1*-Mut and *BRCA1*-Rev cells following ADAR1 silencing using RNA-Seq. This revealed (i) a constitutively elevated ISG signature in SUM149 *BRCA1*-Mut cells in absence of ADAR1 silencing (Supplementary Fig. 8H; LFC = 1.66 for *BRCA1*-Mut vs. *BRCA1*-Rev, *P* < 0.0001), and (ii) a selective upregulation of the ISG signature upon ADAR1 silencing in SUM149 *BRCA1*-Mut cells (Supplementary Fig. 8I; LFC = 0.34 for siADAR1 vs. siCTRL in *BRCA1*-Mut, *P* = 0.0029; LFC = 0.16 for siADAR1 vs. siCTRL in *BRCA1*-Rev, *P* = 0.0617, not significant), thereby confirming our previous observations (Fig. 6A–F).

We next directly assessed whether the *BRCA1/2–ADAR1* SL might depend upon cytosolic nucleic acid sensing, and if so, sought to identify which PRR might be involved. To this aim, we conducted co-silencing experiments in which SUM149 *BRCA1*-isogenics were transfected with either *ADAR1* siRNA and a non-targeting control siRNA or with *ADAR1* siRNA and one of a series of siRNAs targeting cytosolic dsRNA sensors (*RIG1, MDA5, LGP2, PKR*) or the non-canonical RNA:DNA hybrid sensor *cGAS*. In these assays, we found that the concomitant silencing of ADAR1 with either RIG1, MDA5, LGP2, PKR or cGAS resulted in a significant reversal of *ADAR1* SL effects in SUM149 *BRCA1*-Mut cells (Fig. 6G, H), while the knockdown of each sensor individually had little effect on cell viability in absence of ADAR1 silencing (Supplementary Fig. 8J-M). If the extent of this rescue was modest with *MDA5* and *RIG1* knockdown, *LGP2, PKR* or *cGAS* knockdown elicited a substantial reversal of the SL, akin to the effect of type I interferon receptor (IFNAR1) silencing, used as a positive control (Fig. 6G, H). Consistent with this, we observed that co-transfection of *LGP2* siRNA was sufficient to abrogate *ADAR1* siRNA-induced phosphorylation of PKR and IRF3, while co-transfection of *RIG1* or *MDA5* siRNA only partially hindered these effects (Fig. 6I). Co-transfection of *PKR* siRNA did not affect *ADAR1* siRNA-induced phosphorylation of IRF3, in line with the notion that PKR is activated by and acts downstream of type I interferon signaling[50]. Of note, silencing of *cGAS* also reduced the expression levels of *RIG1, MDA5,* and *LGP2* while conversely, silencing of either *RIG1, MDA5,* or *LGP2* reduced the expression levels of all three dsRNA sensors (as well as phosphorylation levels of PKR) but not those of cGAS (Fig. 6I). This indicated an interdependence between cytosolic dsRNA, RNA:DNA and DNA sensing pathways as previously described[51], and suggested some level of redundancy among these proteins in mediating the *BRCA1/2–ADAR1* SL effects.

To corroborate this data, we tested whether increased interferon signaling might also directly contribute to the *BRCA1/2–ADAR1* SL. To do so, we evaluated the cytotoxic effects of silencing ADAR1 in presence of non-toxic concentrations of pharmacological inhibitors of the JAK-STAT pathway (JSPi) – namely ruxolitinib (JAK1/2 inhibitor), upadacitinib (JAK1 inhibitor) or deucravacitinib (TYK2 inhibitor), which are commonly used to block downstream interferon signaling[52,53]. JSPi completely reversed *ADAR1* SL effects elicited in *BRCA1*-mutant SUM149 and MEF cells in short-term survival assays, and these effects further extended to the abrogation of ADAR1-dependent dsRNA-sensing PRR upregulation, PKR and IRF3 phosphorylation (Fig. 6J, K; and Supplementary Fig. 8N, 9A–C). To further test the influence of interferon signaling stimulation on the *BRCA1/2–ADAR1* SL, we complemented cell culture media with interferons, and found that non-toxic concentrations of either type I (IFN-α, IFN-β) or type II (IFN-γ) interferons (i) enhanced the cytotoxic effects of ADAR1 silencing in SUM149 *BRCA1*-Mut cells; and (ii) sensitized SUM149 *BRCA1*-Rev cells to ADAR1 silencing (Supplementary Fig. 9D–F). We noted that, compared with IFN-γ, type I interferons caused a more profound sensitization of *BRCA1*-revertant cells, also replicated in *BRCA1*-wildtype MDA-MB-231 cells (Supplementary Fig. 9G–I).

To investigate the role and biological relevance of PKR activation in the *BRCA1/2–ADAR1* SL, we assessed the formation of cytosolic G3BP1 bodies—a marker of stress granules, the prototypical hallmark of ISR—by immunofluorescence in SUM149 *BRCA1*-isogenics. ADAR1 silencing caused a substantial accumulation of cytosolic G3BP1 bodies selectively in *BRCA1*-mutant cells, associated with increased colocalizing PKR foci (Supplementary Fig. 9J–L). Based on this, we reasoned that ISR signaling might play a role in *BRCA1/2–ADAR1* SL, and next assessed the effect of ISRIB—a pharmacological inhibitor of the ISR[54], on the sensitivity of *BRCA1*-mutant cells to ADAR1 silencing. Short-term survival assays revealed a partial rescue of *ADAR1* siRNA-mediated SL effects following exposure to ISRIB in SUM149 *BRCA1*-Mut cells (Supplementary Fig. 9M), in contrast with the complete rescue observed with JSPi. This was associated with an impaired formation of cytosolic G3BP1 bodies and PKR foci (Supplementary Fig. 9N–P), confirming ISRIB inhibitory activity and suggesting a contribution of PKR-driven ISR to the *BRCA1/2–ADAR1* SL.

Since both RH1 overexpression and PRR knockdown reversed the *BRCA1/2–ADAR1* SL, we assessed the contribution of R-loops to PRR-driven innate immune responses. To do this, we replicated western blots of dsRNA sensors, type I interferon and ISR markers in SUM149 *BRCA1*-isogenic cells subjected to (i) ADAR1 silencing and/or (ii) RH1 overexpression and/or (iii) JSPi exposure. Whilst *BRCA1*-mutant cells were unable to activate a type I interferon response upon ADAR1 silencing when exposed to JSPi (Supplementary Fig. 10A) or co-silencing of RIG1 or LGP2 (Supplementary Fig. 10B), RH1 overexpression also counteracted (albeit partially) the upregulation of PRR, type I interferon and integrated stress response markers elicited upon

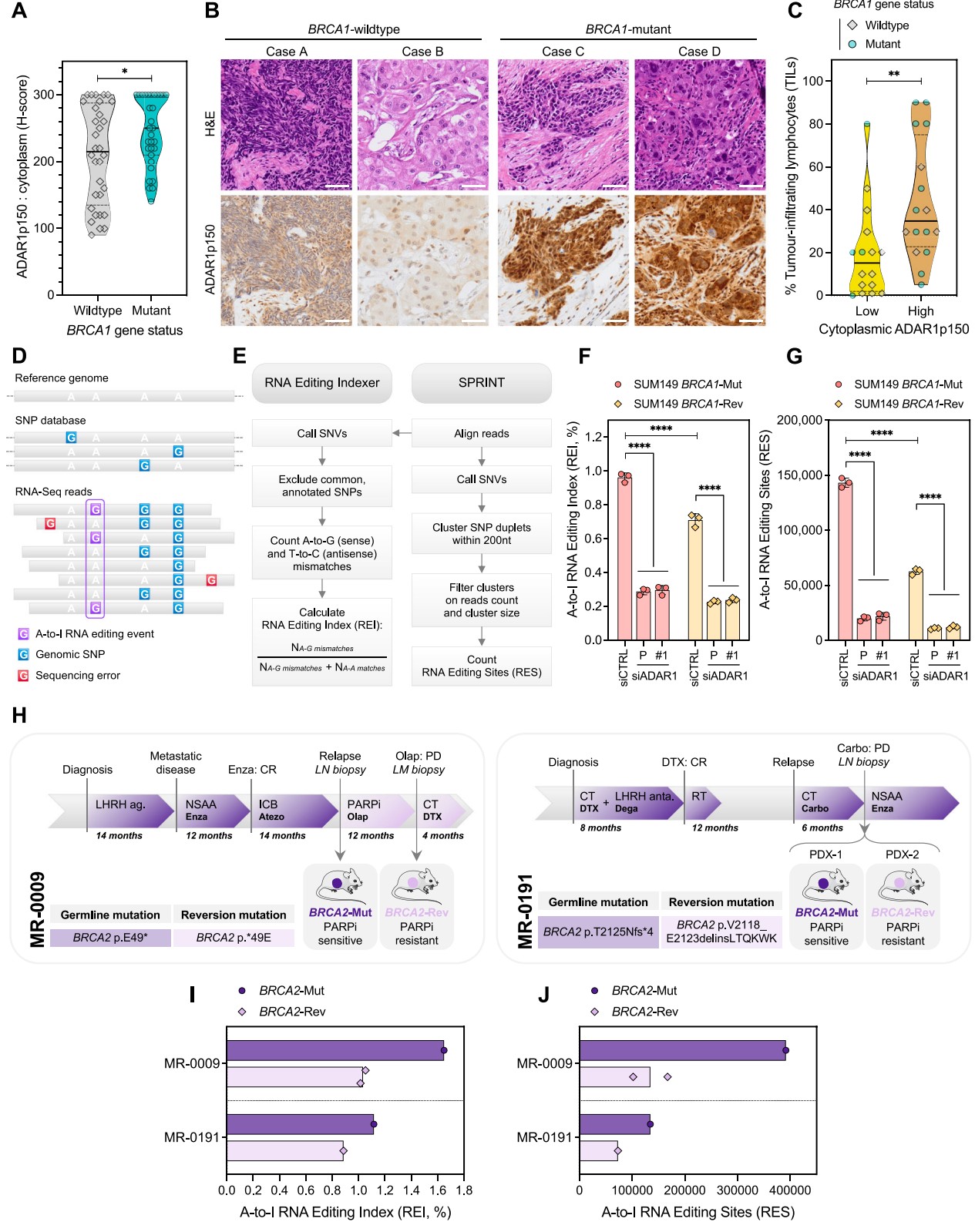

ADAR1 silencing in *BRCA1*-mutant cells (Supplementary Fig. 10A, B). Consistent with this, short-term survival assays conducted in the same conditions revealed that RH1 overexpression and exposure to JSPi both contributed to reverse the cytotoxic effects of *ADAR1* siRNA in SUM149 *BRCA1*-Mut cells (Supplementary Fig. 10C).

Altogether, this data supported the idea that increased R-loop burden elicited by ADAR1 suppression associates with PRR-driven cell-autonomous interferon poisoning in *BRCA*m cells, thereby ultimately causing the *BRCA1/2–ADAR1* SL.

## Discussion

Here, we uncover the defect in the RNA editing enzyme ADAR1 as a novel genetic vulnerability of *BRCA1/2*-mutant (*BRCA*m) cancer cells. Using functional genomics in a panel of diverse *BRCA1/2*-isogenic

**Fig. 3 | BRCA1/2 mutations in patients associate with increased tumor cell ADAR1 expression and activity. A, B** Pathological evaluation (**A**) and representative images (**B**) of ADAR1p150 cytoplasmic expression according to *BRCA1* gene status (*BRCA1*-wildtype vs. *BRCA1*-mutant) in a cohort of 63 treatment-naïve triple-negative breast cancer (TNBC) patients. H-score of ADAR1p150 expression (range, 0–300). Violin plots indicate median, lower and upper quartiles; N = 32 [*BRCA1*-wildtype], N = 31 [*BRCA1*-mutant] values from individual tumor samples, two-tailed Mann-Whitney U test. *P* value, *=0.0359. Hematoxylin and eosin (H&E) and ADAR1p150 staining (magnification, ×20) are shown. Scale bars, 50 µm. **C** Percentage of TILs in TNBC tumors from the cohort described in **A**, according to cytoplasmic ADAR1p150 expression (based on **A**; ADAR1p150-low, lower quartile of H-score; ADAR1p150-high, upper quartile of H-score). Violin plots indicate median, lower and upper quartiles; N = 17 [*BRCA1*-wildtype], N = 15 [*BRCA1*-mutant] values from individual tumor samples, two-tailed Mann-Whitney U test. *P* value, **=0.0043. **D, E** Schematics illustrating the conceptual approach (**D**) and bioinformatic pipelines (**E**) used to evaluate A-to-I editing levels from RNA-Seq data.

**F, G.** A-to-I RNA editing levels displayed as RNA editing index (**F**) or number of RNA editing sites (**G**) in SUM149 *BRCA1*-Mut and *BRCA1*-Rev cells transfected with *ADAR1* siRNA (P, Pool; #1). siCTRL, non-targeting, negative control siRNA. Bar plots indicate mean ± SD; *n* = 3 biological replicates, two-way ANOVA post hoc Tukey's test. *P* values, ****< 0.0001. **H** Schematic of clinical history of the *BRCA2*-mutant and -revertant patient-derived xenografts (PDXs) MR-0009 and MR-0191; arrows indicate times of tumor biopsies for PDX establishment. Duration of each treatment delivered after diagnosis is indicated in months. Details of the corresponding *BRCA2* mutations (germline vs. reversion) are presented to the left. **I, J** A-to-I RNA editing levels displayed as RNA editing index (**I**) or number of RNA editing sites (**J**) in *BRCA2*-mutant and -revertant PDXs MR-0009 and MR-0191. Bar plots indicate mean, where applicable; N = 1 [MR-0009 *BRCA2*-Mut], N = 2 [MR-0009 *BRCA2*-Rev], N = 1 [MR-0191 *BRCA2*-Mut], N = 1 [MR-0191 *BRCA2*-Rev] values from individual tumor samples. Source data are provided as a Source Data file. Elements of panel **H** were provided by Servier Medical Art (https://smart.servier.com/), licensed under CC BY 4.0 (https://creativecommons.org/licenses/by/4.0/).

model systems, we identify a robust and penetrant synthetic lethal (SL) interaction between *BRCA1/2* mutations and the loss of ADAR1, which operates across a variety of molecular backgrounds, cancer histotypes, and species, and also exists reciprocally, in *ADAR1*-mutated systems (Figs. 1, 2). The evidence that ADAR1 is dysregulated in *BRCA*m human tumors and PDXs (Fig. 3) supports the translational significance of this SL. Mechanistically, we show that the *BRCA1/2–ADAR1* SL relies upon R-loop-associated DNA damage (Figs. 4, 5) and the selective activation of an R-loop- and PRR-driven cell-autonomous innate immune response in *BRCA*m cells (Fig. 6), providing the first evidence for autocrine interferon poisoning as a mechanism of *BRCA1/2* SL[46].

We show that when ADAR1 is silenced in *BRCA1*-mutant cells, overexpression of RNase H1—which reduces R-loop burden— antagonizes *ADAR1* SL effects as well as the upregulation of cytosolic nucleic acid sensing PRR, type I interferon and integrated stress response markers, suggesting that R-loops accumulation triggers, directly or indirectly, downstream pattern recognition receptors (PRR) activation and interferon-dependent lethality. Importantly, these findings delineate a distinctive SL mechanism, which differs from those of other *BRCA1/2*-associated SL (including the *BRCA1/2*–PARPi SL) in that it depends on cytosolic nucleic acid sensing PRR activation. In line with the recently described immunogenic potential of R-loops[55–62], our results indeed support a model (Fig. 7) in which unresolved R-loops elicited by the loss of ADAR1 function[4,6,48] activate cytosolic nucleic acid sensing PRR, resulting in a toxic cell-autonomous innate immune response which drives the *BRCA1/2–ADAR1* SL. ADAR1p150 plays an essential role in this SL and could be responsible for the suppression of nuclear R-loops in *BRCA*m cells, consistent with its nucleocytoplasmic distribution[63,64]. Still, some elements of the *BRCA1/2–ADAR1* SL mechanism remain to be explored.

First, the PRR specificity of the *BRCA1/2–ADAR1* SL remains unclear. Our results indicate that LGP2, PKR and cGAS silencing most reproducibly rescue *ADAR1* SL effects, suggesting a contribution of multiple cytosolic nucleic sensing pathways. Whilst cGAS and PKR are known to detect RNA:DNA hybrids similar to those formed at R-loops[55,65], LGP2 has not yet been described to do so, suggesting either that its substrate specificity might be less discrete than expected, or that the activation of some PRR, through crosstalk, causes activation of others. Indeed, considering the interdependence and possible redundancy of these pathways, other PRR might also contribute to these SL effects (e.g., TLR3 or ZBP1[55,66]). Intriguingly, silencing of DHX9 (a PRR which both suppresses cytosolic dsRNA sensing[59,67] and bears a nuclear RNA helicase activity against R-loops[68]) did not elicit *BRCA1*-dependent SL effects in our original screen, suggesting a possible redundancy of DHX9 function (i) at the nuclear level, with that of other helicases for the resolution of R-loops and/or (ii) at the cytosolic level, with that of other PRR for the suppression of dsRNA sensing-mediated responses.

In this context, the exact nature of cytosolic nucleic acid species triggering cell-autonomous interferon poisoning remains to be defined. Although our data point to a central role of R-loops in the mechanism of this SL, multiple cytosolic nucleic sensing PRR—not restricted to those described as putative RNA:DNA hybrid sensors (i.e., cGAS[55,69], TLR3[55], TLR9[70] and NLRP3[71]), are activated by ADAR1 silencing in *BRCA1/2*-mutant cells. Prior work indicates that R-loop-derived cytosolic nucleic acids – including DNA, dsRNA, and RNA:DNA hybrids, can trigger interferon responses via DNA- and dsRNA-sensing PRR[55–62], and conversely, that canonical cytosolic DNA and dsRNA sensors can detect RNA:DNA hybrids[55,65], suggesting a possible contribution of several cytosolic nucleic acid species to the SL. Besides, others have shown that cytosolic dsRNA arises from R-loop-prone genomic regions, including telomeres (via telomeric repeat-containing RNA, TERRA)[4,66,72,73], and micronuclei (via aberrant transcription of chromosomes trapped in micronuclei)[48,74]. These latter findings could be consistent with our observations that an increased R-loop burden also, in the context of the *BRCA1/2–ADAR1* SL, activates dsRNA-sensing PRR, and are further compatible with a possible co-existence of various R-loop-derived immunogenic cytosolic nucleic acids species in response to ADAR1 silencing. Interestingly, our observation that RNase H1 overexpression does not fully abrogate the type I interferon response elicited by ADAR1 silencing in *BRCA1*-mutant cells is in line with the notion that cytosolic RNA:DNA hybrids originate from a subset of long-lived nuclear R-loops that are partially RNase H-resistant[55].

Secondly, the characterization of R-loops as the trigger of ADAR1-dependent SL effects would require further investigation to determine the nature (promoter-paused vs. elongation-associated R-loops[75]), genomic location (promoter proximal, exonic or non-coding regions) and functional context (co-transcriptional vs. DNA repair-associated R-loops at DSBs[76–78] or replication forks[79]) of R-loops involved in the *BRCA1/2–ADAR1* SL. The use of orthogonal methods to visualize (e.g., catalytically-inactive mutant RNase H1 protein) and capture both native (e.g., R-loop CUT&Tag) and *ex cellulo*-isolated R-loops (e.g., DRIP-Seq) would allow a more comprehensive profiling of the R-loop landscape in response to impaired ADAR1 function. Similarly, resolutive methods of detection and analysis of cytosolic RNA:DNA hybrids (e.g., CytoDRIP-Seq[55]) would be needed to better understand how R-loops control the activation status of DNA- and dsRNA-sensing PRR.

Thirdly, complementary mechanisms might also contribute to this SL. These include the antagonism of ADAR1-mediated recoding activity at specific editing sites[80], mitotic failure due to defective ADAR1p150 function in chromosome segregation[81], or ZBP1-dependent necroptosis following defective editing of Alu duplex RNA[82–84]. Yet, the complete abrogation of SL effects upon JAK/STAT pathway inhibition suggests a central role of interferon-dependent, "viral mimicry" responses in mediating the *BRCA1/2–ADAR1* SL

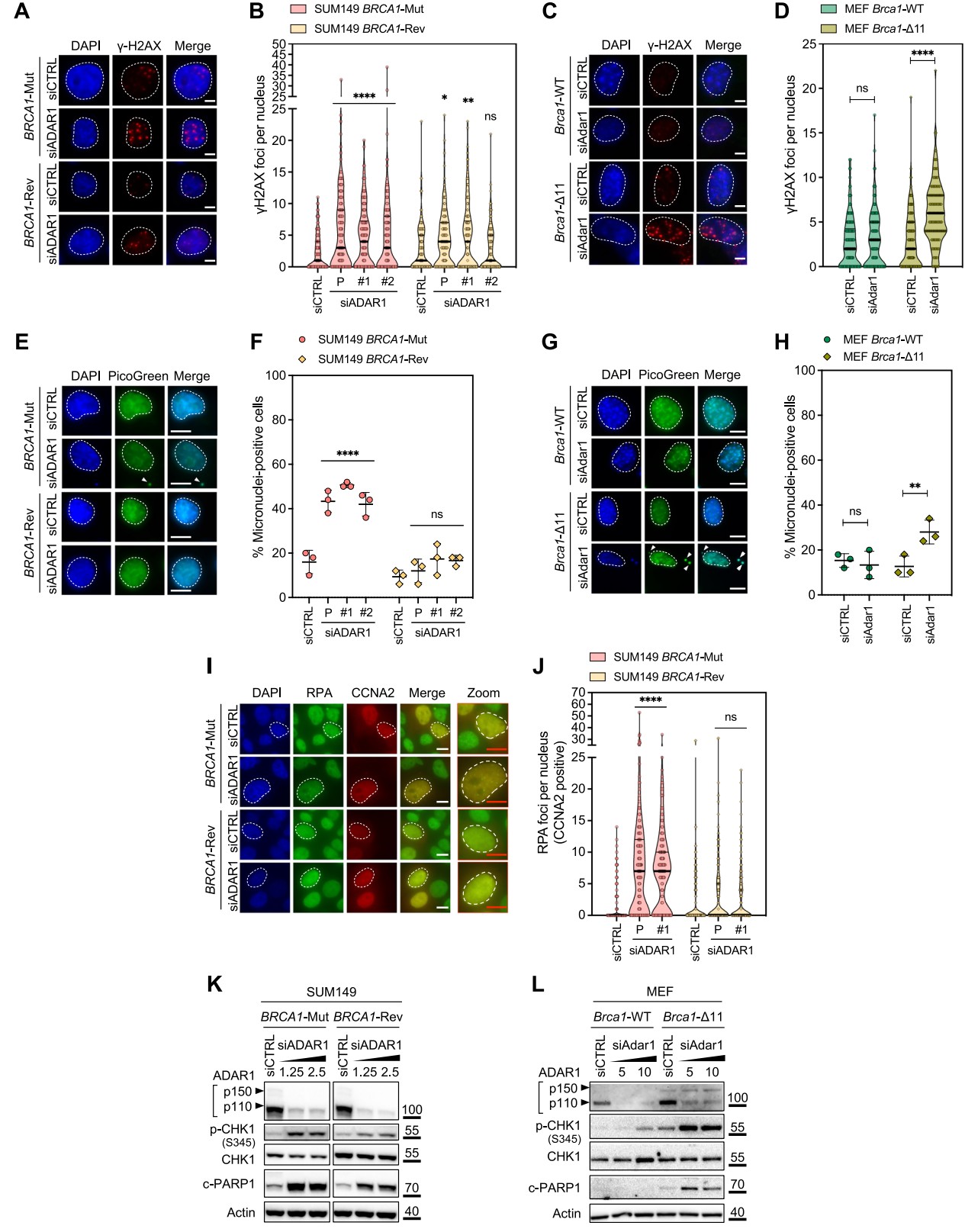

phenotype. Whether these alternative mechanisms may occur synergistically or in a context-specific fashion would deserve further exploration.

Lastly, and most importantly, this SL highlights novel therapeutic opportunities. ADAR1 has previously been proposed as a therapeutic target in TNBC[11–13] and in tumors harboring elevated levels of dsRNA species[14,15] due to enhanced responsiveness to interferon signaling.

However, no clinically-actionable biomarker has thus far been identified that would allow cancer patients to be stratified to receive an ADAR1-targeting agent. Here, we identify *BRCA1/2* mutations—a routinely-used biomarker in oncology—as a robust determinant of sensitivity to ADAR1 inhibition. The multifaceted nature of mechanisms underlying the *BRCA1/2*–ADAR1 SL, which are distinct from those underlying sensitivity or resistance to agents targeting *BRCA1/2*

**Fig. 4 | ADAR1 silencing in *BRCA1*-mutant cells causes DNA damage and replication stress. A–D** Representative images and quantifications of γ-H2AX foci (number of γ-H2AX foci per nucleus) in SUM149 *BRCA1*-Mut and *BRCA1*-Rev cells (**A**, **B**) or MEF *Brca1*-wildtype (WT) and *Brca1*-mutant (Δ11) cells (**C**, **D**) transfected with *ADAR1* siRNA (P, Pool; #1; #2). Violin plots indicate median, lower and upper quartiles; N = 150 values from individual nuclei, representative of *n* = 3 biologically-independent experiments, two-way ANOVA post hoc Dunnett's test (**B**) or Šidák's test (**D**). *P* values, *=0.0101, **=0.0014, ****< 0.0001. **E–H** Representative images and quantifications of micronuclei (percentage of cells harboring >1 micronucleus in the assessed population) in SUM149 *BRCA1*-Mut and *BRCA1*-Rev cells (**E**, **F**) or MEF *Brca1*-wildtype (WT) and *Brca1*-mutant (Δ11) cells (**G**, **H**) transfected with *ADAR1* siRNA (P, Pool; #1; #2). Scatter dot plots indicate mean ± SD; N = 3 values from individual microscopic fields, representative of *n* = 3 biologically-independent

experiments, two-way ANOVA post hoc Dunnett's test (**F**) or Šidák's test (**H**). *P* values, **=0.0099, ****< 0.0001. **I, J** Representative images and quantifications of RPA foci (number of RPA foci per nucleus) in CCNA2-positive SUM149 *BRCA1*-Mut and *BRCA1*-Rev cells transfected with *ADAR1* siRNA (P, Pool; #1). Violin plots indicate median, lower and upper quartiles; N = 150 values from individual nuclei, representative of *n* = 3 biologically-independent experiments, two-way ANOVA post hoc Dunnett's test. *P* values, ****<0.0001. **K, L** Western blot of SUM149 *BRCA1*-Mut and *BRCA1*-Rev cells (**K**) or MEF *Brca1*-wildtype (WT) and *Brca1*-mutant (Δ11) cells (**L**) transfected with a concentration range (nM) of *ADAR1* siRNA. Data representative of *n* = 2 biologically-independent experiments. siCTRL, non-targeting, negative control siRNA; siPLK1, *PLK1*-targeting, positive control siRNA. Source data are provided as a Source Data file.

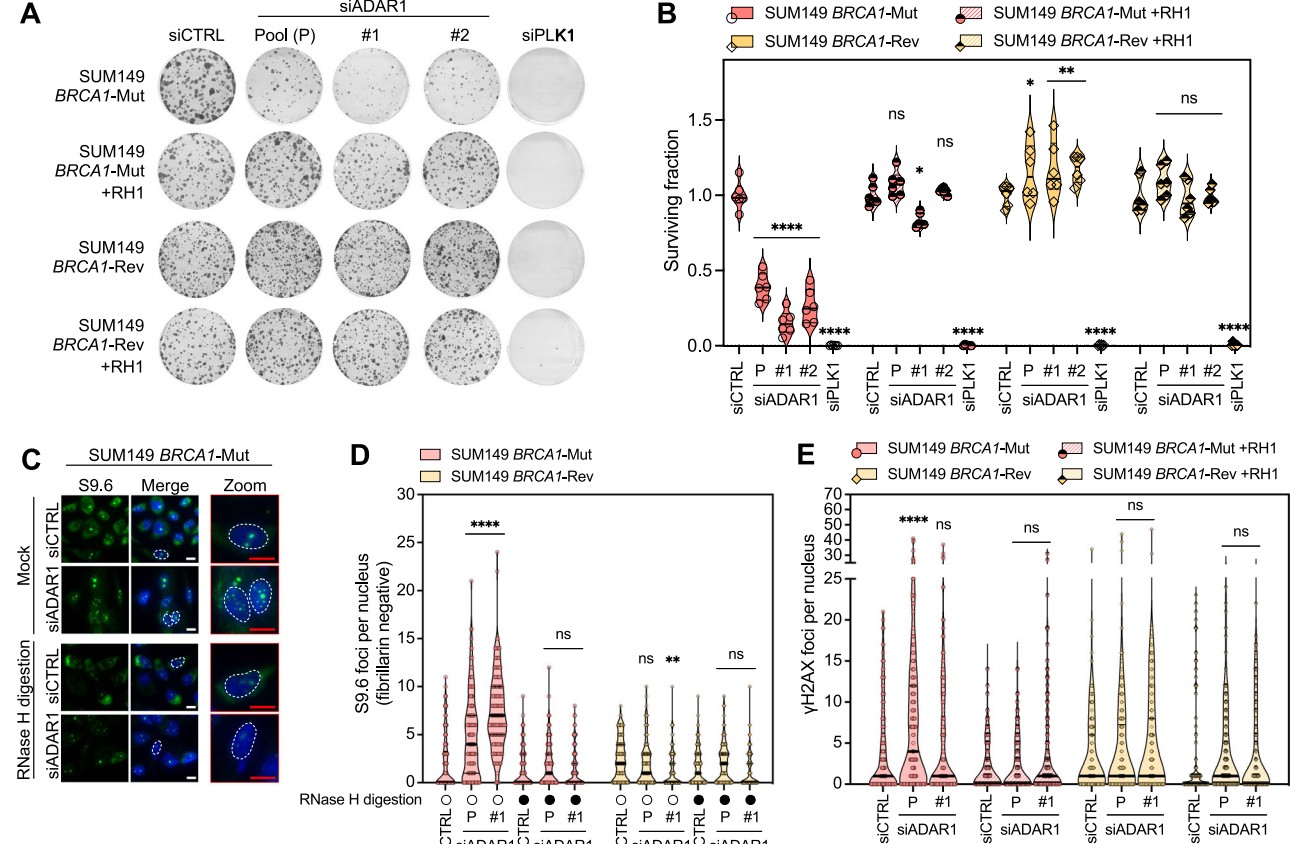

**Fig. 5 | ADAR1 silencing in *BRCA1*-mutant cells causes R-loop-dependent synthetic lethality. A, B** Clonogenic survival of SUM149 *BRCA1*-Mut and *BRCA1*-Rev cells transfected with *ADAR1* siRNA (P, Pool; #1; #2) in the context of exogenous overexpression of RNase H1 (RH1). Violin plots indicate median, lower and upper quartiles; N = 6 values from individual wells, representative of *n* = 3 biologically-independent experiments, two-way ANOVA post hoc Dunnett's test. *P* values, *[*BRCA1*-Mut +RH1, siADAR1-#1] = 0.0109, *[*BRCA1*-Rev, siADAR1-P] = 0.019, **[*BRCA1*-Rev, siADAR1-#1] = 0.0084, **[*BRCA1*-Rev, siADAR1-#2] = 0.0071, ****<0.0001. **C, D** Representative images and quantifications of R-loops (number of fibrillarin-negative S9.6 foci per nucleus) in SUM149 *BRCA1*-Mut and *BRCA1*-Rev cells transfected with *ADAR1* siRNA (P, Pool; #1). Violin plots indicate median, lower

and upper quartiles; N = 150 values from individual nuclei, representative of *n* = 3 biologically-independent experiments, two-way ANOVA post hoc Tukey's test. *P* values, **=0.0032, ****< 0.0001. **E** Quantification of γ-H2AX foci in SUM149 *BRCA1*-Mut and *BRCA1*-Rev cells transfected with *ADAR1* siRNA (P, Pool; #1) in the context of exogenous overexpression of RNase H1 (RH1). Violin plots indicate median, lower and upper quartiles; N = 150 values from individual nuclei, representative of *n* = 3 biologically-independent experiments, two-way ANOVA post hoc Dunnett's test. *P* value, ****< 0.0001. siCTRL, non-targeting, negative control siRNA; siPLK1, *PLK1*-targeting, positive control siRNA. Source data are provided as a Source Data file.

defects, such as PARPi, chemotherapy or other DDR inhibitors (DDRi), suggests that targeting ADAR1 in platinum- or PARPi-resistant settings might deserve evaluation. Similarly, combining an ADAR1-targeting agent with DDRi in *BRCA*m cancers, may be complementary to the currently-evaluated DDRi-DDRi combinations, whose tolerability is challenging. Also, whether this SL could be extended to other HR defects, or tumors that harbor a "BRCAness" phenotype, remains to be explored. Still, despite recent attempts to develop ADAR1-targeting

agents[85,86], very few promising preclinical candidates exist[87] and poorly-selective adenosine analogs virtually remain the only commercially-available preclinical compounds[88], urging the development of potent, selective ADAR1 small-molecule inhibitors or degraders.

In conclusion, we describe a genetic interaction between *BRCA1/2* and *ADAR1* that opens novel perspectives towards the development of ADAR1-targeting approaches in immuno-oncology. Once drug-like

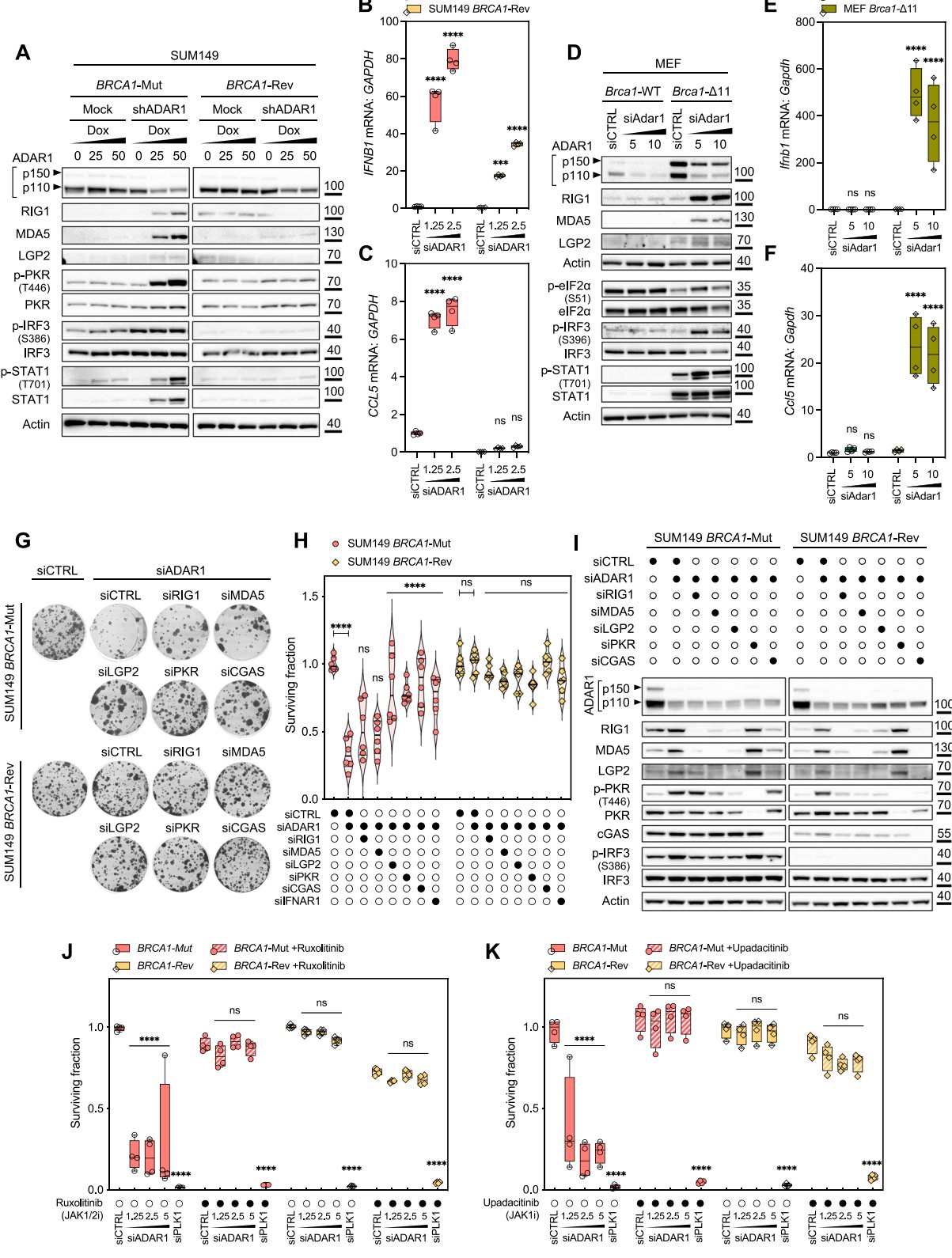

ADAR1 inhibitors are discovered, these could readily be assessed in clinical trials including *BRCA*m patients.

## Methods

This study complies with all relevant ethical regulations, as detailed below. A complete list of all antibodies, RNAi, CRISPR and RT-qPCR reagents is available in Supplementary Information.

### Clinical specimens

Use of breast cancer samples was approved by the institutional ethics review board at Gustave Roussy (CSET, *Commission Scientifique des Essais Thérapeutiques*), in accordance with the Declaration of Helsinki. Informed consent was obtained from all subjects for the use of their tumor tissue as part of this study. A total of 63 TNBC surgical specimens were collected between January 2005 and September 2023.

**Fig. 6 | The *BRCA1/2*–*ADAR1* synthetic lethality requires pattern recognition receptors and interferon signaling. A** Western blot of SUM149 *BRCA1*-Mut and *BRCA1*-Rev cells transduced with a doxycycline-inducible *ADAR1*-targeting shRNA and exposed to a titration of doxycycline (ng/mL). Data representative of *n* = 2 biologically-independent experiments. **B, C** RT-qPCR of *IFNB1* (**B**) and *CCL5* (**C**) mRNAs in SUM149 *BRCA1*-Mut and *BRCA1*-Rev cells transfected with a concentration range (nM) of *ADAR1* siRNA. *IFNB1* and *CCL5* mRNAs were analyzed separately relative to *GAPDH*. Box-and-whiskers show arbitrary units of gene expression normalized to the *BRCA1*-mutant siCTRL condition; N = 4 values from individual measurements, representative of *n* = 3 biologically-independent experiments, two-way ANOVA post hoc Dunnett's test. *P* values, ***[*IFNB1*, *BRCA1*-Rev, siADAR1 1.25 nM] =0.0002, ****< 0.0001. **D** Western blot of MEF *Brca1*-wildtype (WT) and *Brca1*-mutant (Δ11) cells transfected with a concentration range (nM) of *Adar1* siRNA. Data representative of *n* = 2 biologically-independent experiments. **E, F** RT-qPCR of *Ifnb1* (**E**) and *Ccl5* (**F**) mRNAs in MEF *Brca1*-wildtype (WT) and *Brca1*-mutant (Δ11) cells transfected with a concentration range (nM) of *Adar1* siRNA. Data presented as in

(**B, C**). *P* values, ****< 0.0001. **G, H** Clonogenic survival of SUM149 *BRCA1*-Mut and *BRCA1*-Rev cells subjected to co-transfection with *ADAR1* siRNA and one of a series of siRNAs targeting pattern recognition receptors. Violin plots indicate median, lower and upper quartiles; N = 6 values from individual wells, representative of *n* = 3 biologically-independent experiments, two-way ANOVA post hoc Dunnett's test. *P* values, ****< 0.0001. **I** Western blot of SUM149 *BRCA1*-Mut and *BRCA1*-Rev cells subjected to co-transfection with *ADAR1* siRNA and one of a series of siRNAs targeting pattern recognition receptors. Data representative of *n* = 3 biologically-independent experiments. **J, K** Cell survival of SUM149 *BRCA1*-Mut and *BRCA1*-Rev cells transfected with a concentration range (nM) of *ADAR1* siRNA in the context of exposure to the JAK/STAT pathway inhibitors (JSPi) ruxolitinib (**J**; 10 μM) or upadacitinib (**K**; 32 μM). Box-and-whiskers indicate median, lower and upper quartiles, and the min to max range; N = 4 values from individual wells, representative of *n* = 3 biologically-independent experiments, two-way ANOVA post hoc Dunnett's test. *P* values, ****< 0.0001. siCTRL, non-targeting, negative control siRNA; siPLK1, *PLK1*-targeting, positive control siRNA. Source data are provided as a Source Data file.

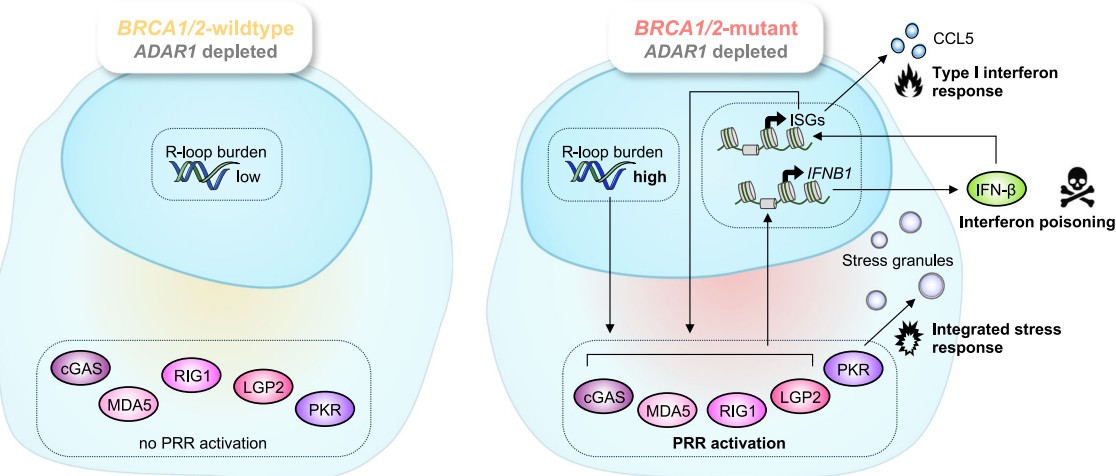

**Fig. 7 | ADAR1 protects *BRCA1/2*-mutant cancer cells against fatal autocrine interferon poisoning.** A model for the proposed mechanism driving sensitivity of *BRCA1/2*-mutant cancers to ADAR1 inhibition. Elements of this figure were provided by Servier Medical Art (https://smart.servier.com/) and BioRender (https://www.biorender.com/), licensed under CC BY 4.0 (https://creativecommons.org/licenses/by/4.0/).

Samples were routinely fixed immediately after surgery in 10% formalin for 24 h at room temperature. After fixation, samples were dehydrated, incubated in xylene, infiltrated with paraffin, and finally embedded in paraffin.

### Animal studies
All animal experiments were conducted in compliance with institutional animal protocols at Inserm, approved by DDPP Val-de-Marne, France, under license number F 94-043-013. AB strain of zebrafish was used in this study. Zebrafish embryos were staged and cared for according to standard protocols[89].

### Cell lines
*BRCA1*-isogenic SUM149 cells (described in ref. 22 and[90]; original source: Asterand) were cultured under normal growth conditions (37 °C, 5% CO₂) in Ham's F-12 medium (Thermo Fisher, #21765037) supplemented with 10% (v/v) heat-inactivated fetal bovine serum (HI-FBS; Sigma Aldrich, #F7524), 5 μg/mL human insulin (Santa Cruz, #sc-360248), 1 μg/mL hydrocortisone (Sigma Aldrich, #H4001) and 1% (v/v) penicillin/streptomycin (P/S; Thermo Fisher, #15140122). MDA-MB-436, MDA-MB-231, CAL51, CAL120, HCC1937 and Hs578T (source: ATCC) were maintained under normal growth conditions in Roswell Park Memorial Institute-1640 (RPMI-1640; Thermo Fisher, #61870044) or high-glucose Dulbecco's Modified Eagle Medium (DMEM; Thermo Fisher, #31966047) supplemented with 10% (v/v) HI-

FBS and 1% (v/v) P/S. *BRCA1*-isogenic TP53⁻/⁻ RPE1 cells (described in ref. 24; gift from D. Durocher) and *BRCA2*-isogenic DLD1 cells (source: Horizon Discovery) were maintained under normal growth conditions in DMEM supplemented with 10% (v/v) HI-FBS and 1% (v/v) P/S. *Brca1*-isogenic mouse embryonic fibroblasts (MEF; generated as previously described[23]) and *Brca1/2*-isogenic 4T1 cells (described in ref. 25; gift from R. Samstein) were maintained under normal growth conditions in RPMI-1640 supplemented with 10% (v/v) HI-FBS and 1% (v/v) P/S. *Brca1*-isogenic ID8 cells (described in ref. 26; gift from I. McNeish) were maintained under normal growth conditions in DMEM supplemented with 10% (v/v) HI-FBS, 1X insulin-transferrin-selenium (ITS; Thermo Fisher, #41400045) and 1% (v/v) P/S. *ADAR1*-isogenic HEK293T (described in ref. 28; gift from C. Rice) were maintained under normal growth conditions in DMEM supplemented with 10% (v/v) HI-FBS and 1% (v/v) P/S. Mycoplasma testing was performed monthly using the MycoAlert Mycoplasma Detection Kit (Lonza, #LT07-318), and all cell lines were short-tandem-repeat (STR) typed using the GenePrint® 10 System (Promega, #B9510) to confirm identity prior to the study.

### Antibodies
Primary antibodies were used for western blotting, immunofluorescence and immunohistochemistry (see Supplementary Table 1 for details). The following secondary antibodies (ThermoFisher) were used for immunofluorescence: Anti-Mouse Alexa Fluor 647 (#A-21422),

Anti-Rabbit Alexa Fluor 488 (#A-11034), Anti-Rabbit Alexa Fluor 647 (#A-21244), Anti-Mouse Alexa Fluor 488 (#A-11001).

## Drugs and chemicals

The JAK/STAT pathway inhibitors ruxolitinib (INCB-18424; JAK1/2i), upadacitinib (ABT-494; JAK1i) and deucravacitinib (BMS-986165, TYK2i), and the integrated stress response inhibitor ISRIB were purchased from MedChemExpress. The PARPi talazoparib (BMN-673) and olaparib (AZD2281) were purchased from SelleckChem. Inhibitor stock solutions were prepared in DMSO and stored in aliquots at -80 °C.

## RNAi and siRNA transfection

To perform siRNA-mediated gene silencing, cells were either (i) forward-transfected i.e., seeded at 60% density in 6-well plate format and transfected 24 h later (for immunofluorescence, western blotting and RT-qPCR); or (ii) reverse-transfected i.e., seeded and transfected in suspension at seeding (in 96-well plate format for short-term survival assays; in 6-well plate format for colony formation assays), with the indicated siRNA using Lipofectamine RNAiMAX (Thermo Fisher, #13778150) and Opti-MEM™ (Thermo Fisher, #31985062) according to manufacturers' instructions. siRNAs targeting the following genes were used: *ADAR1*, *BRCA1*, *BRCA2*, *CGAS*, *IFNAR1*, *LGP2*, *MDA5*, *PKR*, and *RIG1* (see Supplementary Table 2 for details). In siRNA co-transfection experiments, equimolar concentrations of each siRNA were used in all conditions. The transfection efficiency was monitored by assessment of cell growth inhibition (>95%) in pools of cells concomitantly transfected with a positive control, *PLK1*-targeting siRNA. Transcriptional silencing efficiency was systematically assessed via evaluation of protein expression by western blotting on pools of concomitantly transfected cells, 72 h after transfection.

## siRNA synthetic lethal screen

SUM149 *BRCA1*-isogenic cells were reverse-transfected with a library of 22 siRNAs, including three non-targeting control siRNAs (siCTRL#1, #2, #3), one lethal control siRNA (siPLK1), and 18 siRNAs targeting genes of interest. The custom library (purchased from Horizon Discovery) was arrayed in 96-well plates and transfections were conducted in quadruplicates, using Lipofectamine RNAiMAX (Thermo Fisher, #13778150) and Opti-MEM™ (Thermo Fisher, #31985062) according to manufacturers' instructions. The transfection media was removed one day after transfection, and after six subsequent days of continuous culture, cell viability was determined by use of CellTiter-Glo® luminescent reagent (Promega, #G7573). Surviving fractions elicited by each siRNA were calculated compared to the median of non-targeting control siRNA conditions.

## CRISPR-Cas9 gene editing and crRNA transfection

To perform CRISPR-Cas9-mediated inactivation of *ADAR1*, cells were reverse-transfected in 6-well plate format with the indicated Edit-R crRNA (Horizon Discovery; see Supplementary Table 3 for details), Edit-R tracrRNA (Horizon Discovery, #U-002005-50) and recombinant Cas9 protein (15 pmol/well), using Lipofectamine CRISPRmax (Thermo Fisher, #CMAX00015) and Opti-MEM™ (Thermo Fisher, #31985062) according to manufacturers' instructions. The transfection efficiency was monitored by assessment of cell growth inhibition (>95%) in pools of cells concomitantly transfected with a positive control, *PLK1*-targeting crRNA. Genetic inactivation efficiency was systematically assessed via evaluation of protein expression by western blotting on pools of concomitantly transfected cells, 48 h after transfection.

## Virus production

Lentiviral particles were generated by transfecting HEK293T cells, using Lipofectamine 2000 (Thermo Fisher, #11668019), with the expression plasmid of interest and the psPAX2 (Addgene, #12260) and pMD2.G (Addgene, #12259) packaging vectors at a ratio of 1:1:1,

respectively. Viral supernatants were collected 3 and 4 days after transfection and filtered through a 0.45 μm strainer. Viral particles were precipitated by incubation of the supernatants O/N at 4 °C with 1X PEG-it Virus Precipitation Solution (System Bioscience, #LV810A-1). The supernatants were then centrifuged at 1500 g for 30 min, and precipitates were resuspended in 2 mL culture media. The resuspended particles were then immediately used for transduction (250 μL per 60 cm² petri dish) or kept at -80 °C for long-term storage.

## Plasmid constructions and stable lentiviral transduction

To generate stable RH1-expressing cells in the SUM149 *BRCA1*-isogenic, MDA-MB-436 and MDA-MB-231 models, cells were transfected with ppyCAG-RNaseH1-V5 plasmid (Addgene, #111906) with Lipofectamine 2000 (Thermo Fisher, #11668019) according to manufacturer's instructions. Stable pools of transfectants were generated by selection with hygromycin B (SUM149, 100 μg/mL; MDA-MB-436, 200 μg/mL; MDA-MB-231, 400 μg/mL) and the resultant selected populations were submitted to clonal isolation using the limiting dilution method. Clones were recovered and profiled for RNase H1 or V5 expression by western blot.

To generate stable doxycycline-inducible shADAR1-expressing cells in the SUM149 *BRCA1*-isogenic model, cells were transduced with a pLKO-Tet-On lentiviral vector (Addgene, #21915) engineered to carry the *ADAR1*-targeting shRNA sequence 3'-TTACCAAGGCCTGAGATA-TAACTCGAGTTATATCTCAGGCCTTGGTAA-5'. Stable pools of transductants were generated by selection with puromycin (1 μg/mL), and the resultant selected populations were submitted to clonal isolation using the limiting dilution method. Clones were recovered and profiled for ADAR1 expression by western blot following exposure to doxycycline (100 ng/mL).

## Plasmid constructions and transient transfection of ADAR1 mutants

Mutated variants of ADAR1 were generated by site-directed mutagenesis using the Phusion™ Site-Directed Mutagenesis kit (Thermo Fisher, #F541) from plasmids containing wildtype ADAR1p150 (Addgene, #117927) and ADAR1p110 (Addgene, #117928) cDNAs. Catalytically-inactive, RNA editing-defective G1007R[31] (p150-ΔCD) and Z-DNA binding-defective P193A[31] (p150-ΔZα) mutants of ADAR1p150 were generated.

Short-term survival assays were performed in 96-well plates. On day 0, exponentially-growing SUM149 *BRCA1*-isogenic cells were seeded at a density of 15,000 cells per well and reverse-transfected in quadruplicates using Lipofectamine 3000 (Thermo Fisher, #L3000015) and Opti-MEM™ (Thermo Fisher, #31985062), with: (i) 100 pg/mL of the indicated wildtype or mutant ADAR1p110 or ADAR1p150 plasmid constructs, or the corresponding empty vector (Addgene, #117926); and (ii) 5 nM of control, non-targeting siRNA (siCTRL), or an individual siRNA targeting ADAR1 3'UTR. On day 1, the media was removed and replaced with fresh media. On day 6, cell viability was determined by use of CellTiter-Glo® luminescent reagent (Promega, #G7573). Surviving fractions were calculated compared to the non-targeting control siRNA conditions.

## Cell-based assays

**Cell survival assays.** Short-term survival assays were performed in 96-well plates. On day 0, exponentially-growing cells were seeded at a density of 15-25,000 cells per well and reverse-transfected in quadruplicates with the indicated siRNAs using Lipofectamine RNAiMAX (Thermo Fisher, #13778150) and Opti-MEM™ (Thermo Fisher, #31985062) according to manufacturers' instructions. On day 1, the media was removed and replaced with fresh media (or fresh media containing the indicated small-molecule inhibitor, as appropriate, in the case of exposure to JSPi and ISRIB). On day 6, cell viability was determined by use of CellTiter-Glo® luminescent reagent (Promega,

#G7573). Surviving fractions were calculated compared to the non-targeting control siRNA condition.

**Colony-formation assays.** Clonogenic assays were performed in 6-well plates. On day 0, exponentially-growing cells were seeded at a density of 400-600,000 cells per well and reverse-transfected with the indicated siRNAs using Lipofectamine RNAiMAX (Thermo Fisher, #13778150) and Opti-MEM™ (Thermo Fisher, #31985062) according to manufacturers' instructions. On day 1, cells were trypsinized, counted, and re-seeded in triplicates at a density of 1,500 cells per well in 6-well plates. On day 10-12, cells were fixed with 0.5% crystal violet in methanol for 20 min Colonies of > 50 cells were counted manually and surviving fractions were calculated compared to the non-targeting control siRNA condition.

**Cell proliferation assays.** Short-term proliferation assays were performed in 96-well plates, in the same conditions as short-term survival assays. Cell proliferation was measured by use of an Incucyte® SX5 (Sartorius) over a period of 6–10 days after transfection. Growth curves were plotted for each siRNA condition.

## Immunoblotting

Cells were lysed in RIPA lysis and extraction buffer (Thermo Fisher, #89900) supplemented with 0.5% phenylmethylsulfonyl fluoride (PMSF; Sigma Aldrich, #93482) and 1% Halt™ protease and phosphatase inhibitor cocktail (Thermo Fisher, #78444). Lysates were generated on ice, and centrifuged 10 min at 16,900 g prior to supernatant collection. Quantification of total protein in supernatants was performed using the Pierce™ BCA protein assay (Thermo Fisher, #23225), and equal amounts of each sample were loaded into NuPAGE™ 4-12% Bis-Tris (Invitrogen, #NP0335BOX) or 3-8% Tris-Acetate precast gels (Invitrogen, #EA0378BOX) and subjected to electrophoresis using NuPAGE™ MOPS SDS (Invitrogen, #NP000102) or Tris-Acetate SDS running buffers (Invitrogen, #LA00401), respectively. After migration, proteins were transferred to a nitrocellulose membrane (iBlot NC Regular Stacks, Invitrogen, #IB23001). 5% bovine serum albumin (BSA; Sigma Aldrich, #A7990C) in Tris-buffered saline (TBS; Euromedex, #ET220-B) supplemented with 0.1% Tween-20 (TBS-T 0.1%; Sigma Aldrich, #P7949) was used to block the membranes at room temperature (RT) for 1 h. Primary antibodies were diluted in 5% BSA in TBS-T 0.1%, and incubated at 4 °C overnight (O/N). The next day, the membranes were washed three times with TBS-T 0.1% for 10 min, followed by incubation with horseradish-peroxidase-conjugated secondary antibodies at RT for 1 h, in 5% milk in TBS-T 0.1%. The membranes were washed again three times with TBS-T 0.1%, and incubated with Clarity ECL substrate (Biorad, #1705060) or Clarity Max ECL substrate (Biorad, #1705062). The membranes were then imaged with a BioRad ChemiDoc XRS+ chemiluminescent detection system.

## Immunofluorescence

Immunofluorescence assays were performed in 6-well plates. On day 0, exponentially-growing cells were seeded in triplicates at a density 200-350,000 cells per well on 13 mm coverslips. On day 1, cells were forward-transfected with the indicated siRNAs using Lipofectamine RNAiMAX (Thermo Fisher, #13778150) and Opti-MEM™ (Thermo Fisher, #31985062) according to manufacturers' instructions. 6-8 h after transfection, the media was removed and replaced with fresh media (or fresh media containing the indicated small-molecule inhibitor, as appropriate). On day 4, cells were either (i) for γ-H2AX, RAD51, RPA, CCNA2, G3BP1 and PKR immunofluorescence, fixed in 4% (v/v) paraformaldehyde (PFA; Euromedex, EM-15710) in PBS for 10 min at RT, washed twice with PBS, and permeabilized with 0.5% (v/v) Triton X-100 (Sigma Aldrich, #X100) in PBS for 10 min; or (ii) for R-loop (S9.6) and fibrillarin immunofluorescence, fixed in ice-cold 100% methanol

for 7 min at -20 °C, washed twice with PBS, quenched with 0.1 M glycine (made fresh) for 10 min at RT, washed again twice with PBS, and permeabilized with 0.5% (v/v) Triton X-100 in PBS for 7 min For R-loop (S9.6) and fibrillarin immunofluorescence, RNase H digestion was performed by incubating coverslips with RNase H (New England Biolabs, #M0297) diluted 1:50 in 1X RNase H buffer (New England Biolabs, #B0297) for 5 h at 37 °C. After two additional washes with PBS, cells were blocked with 2% (w/v) BSA, 1% (v/v) FBS in PBS (IFF) for 1 h at RT, followed by incubation with the indicated primary antibodies in IFF at 4 °C overnight. The next day, cells were washed three times with PBS for 10 min, and incubated with the indicated secondary antibodies, DAPI (Thermo Fisher, #62248) and PicoGreen® (Thermo Fisher, #P7581; only when assessing micronuclei) in IFF for 1 h at RT. After three additional washes with PBS for 10 min, coverslips were dried and mounted on microscope slides using Fluoromount™ mounting medium (Sigma Aldrich, #F4680). Slides were imaged at 40× on a digital slide scanner (Hamamatsu).

## Analysis of immunofluorescence images

Quantification of the number of micronuclei, γ-H2AX, RAD51, S9.6, RPA, G3BP1, and PKR foci was performed manually under identical microscopy settings between samples, using the OlyVIA image viewer software (Olympus) for visualization. Three independent fields comprising a minimum of 150 cells were used. For quantification of R-loops (S9.6 foci), only nuclear foci outside nucleoli (fibrillarin-positive bodies) were counted. For quantification of RPA foci, only CCNA2-positive cells were used in the analysis.

## Immunohistochemistry

For each patient sample, a single representative formalin-fixed paraffin embedded (FFPE) block was selected for the study. FFPE blocks were sectioned (4 µm thick) on a RM2245 microtome (Leica Biosystems) and placed onto histological TOMO® adhesion microscope slides (VWR, #10748-166). ADAR1 and ADARp150 automated immunohistochemistry staining was performed using a BOND RX automated research stainer (Leica Biosystems). After deparaffinization with BOND™ Dewax solution (Leica, #AR9222), epitope retrieval was performed through incubation in BOND™ Epitope Retrieval solution 2 (Leica, #AR9640) at 100 °C, pH 9 for 20 min, and endogenous peroxidase activity was inhibited by treatment with the Peroxide Block reagent of the BOND™ Polymer Refine Detection kit (Leica, #DS9800) for 5 min The slides were then incubated with primary antibodies for 30 min at RT (ADAR1, 1:700; ADARp150, 1:1,000). Detection was performed with a secondary antibody and subsequently revealed with the 3,3'-diaminobenzidine tetrahydrochloride (DAB) chromogenic substrate of the BOND™ Polymer Refine Detection kit (Leica, #DS9800). Counterstaining was performed by incubation with the hematoxylin reagent of the BOND™ Polymer Refine Detection kit (Leica, #DS9800) for 5 min Hematoxylin and eosin (H&E) automated staining was performed using a GEMINI AS (MM France). Tissue sections were deparaffinized in xylene and rehydrated by incubation in serial 100% ethanol baths. The slides were incubated with haemalun for 4.5 min and then with eosin for 2.5 min Dehydration was carried out by 3 successive baths of 100% ethanol followed by xylene. Coverslips were mounted on microscope slides using PERTEX® mounting medium (VWR, #LEIC811).

## Pathological assessment of TNBC tumors

Archival samples from treatment-naïve TNBC surgical specimen or biopsies were used to build a cohort of 63 cases (*BRCA1*-wildtype, N = 32; *BRCA1*-mutant, N = 31). Blinded histopathological assessment of ADAR1p150 and tumor-infiltrating lymphocytes (TILs) was performed by a senior breast pathologist. For ADAR1p150, expression was evaluated in the nucleus and cytoplasm of tumor cells using an H-score (percentage of stained tumor cells multiplied by each intensity from 0

to 3 +, value from 0 to 300). TILs density was quantified using morphological evaluation of lymphocytes based on a standardized methodology for breast cancer[91].

## RT-qPCR

RNA was extracted from cells using the RNeasy Mini kit (Qiagen, #74104). DNA contamination was removed using the RNase-Free DNase Set (Qiagen, #79256). RNA was then quantified using Nano-Drop™ 2000 spectrophotometer (Thermo Fisher) and diluted to equal concentrations across all samples. Reverse-transcription was performed using the SuperScript VILO cDNA Synthesis kit (Thermo Fisher, #11754250), following the manufacturer's protocol. The qPCR reactions were performed in 384-well plates with TaqMan™ Fast Advanced Master Mix (Thermo Fisher, # 4444963), and samples were analyzed using a ViiA7 real-time PCR system (Applied Biosystems). Probes against the following transcripts were used (see Supplementary Table 4 for details): *CCL5*, *IFNB1*, *IFI44*, *IFIT1*, *MX1*. Results were normalized to *GAPDH*.

## RNA-Seq

RNA was extracted from cells using the RNeasy Mini kit (Qiagen, #74104). DNA contamination was removed using the RNase-Free DNase Set (Qiagen, #79256). Initial quality control and quantification of the RNA material was performed using Qubit RNA HS Assay kit (Thermo Fisher, #Q32852) on a Qubit 2.0 Fluorometer (Thermo Fisher). RNA degradation was determined through evaluation of the RIN, using RNA 6000 Pico kit (Agilent, #5067-1513) on an Agilent 2100 Bioanalyzer System (Agilent). rRNA depletion was conducted using the Illumina® TruSeq Stranded Total RNA Library Prep (Human/Mouse/Rat), and the rRNA-free residue was cleaned up by ethanol precipitation. Library preparation for sequencing was carried out using the NGS Stranded RNA Library Prep Set (Novogene) on the rRNA-depleted RNA, as per manufacturer's instructions. Briefly, after fragmentation, first-strand cDNA synthesis was performed using random hexamer primers, followed by second-strand cDNA synthesis in which dUTPs were replaced with dTTPs in the reaction buffer. Subsequently, the cDNA fragments were subjected to end repair, A-tailing, adapter ligation, size selection, USER enzyme digestion, amplification, and purification. The completed directional libraries were then quantified with Qubit and qPCR, and checked for size distribution detection using RNA 6000 Pico kit (Agilent, #5067-1513) on an Agilent 2100 Bioanalyzer System (Agilent). Quantified libraries were finally pooled and sequenced on a NovaSeq 6000 S4 (Illumina) using a 150 bp paired-end (PE150) sequencing strategy.

Fastq files were both trimmed and controlled with Fastp[92], cutting both front and tail on a window of size 6 and a minimum quality of 10. Reads shorter than 15 bases were taken out, reads with more than 50% of unqualified bases were discarded, and overrepresentation analysis was turned on.

## RNA editing analyses

RNA editing analysis was performed by SPRINT[44] and RNAEditingIndex[43]. SPRINT used cleaned fastq files from Fastp, and each edition event was counted from bed output files with R. Bam files from SPRINT were sorted by Samtools[93] then used by RNAEditingIndex with default parameters to compute REI scores.

## Interferon-stimulated genes (ISG) signature

ISG signature genes were retrieved from Extended Data Fig. 1a of Liu et al.[11]. The ISG signature was evaluated using RNA-Seq data from SUM149 *BRCA1*-isogenic cells and GSVA (version 2.0.1)[94], with the option "kcdf" set to "Poisson". Statistical tests between conditions were performed by limma (version 3.62.1)[95] and heatmap graphs were computed using the ComplexHeatmap R package (version 2.22.0)[96,97].

## Zebrafish studies

**Morpholino (MO) injections.** Knockdown of *brca2* and *adar1* was performed using established translation-blocking antisense *brca2* and *adar1* morpholinos (MO) with the sequences 3'-TTTCAAA-CATGCTGCCATGACTGTG-5'[98] and 3'-TCCCTCCTCTACCTCTGCTCA-TAGC-5'[36], respectively. A standard, widely-used negative control MO was used as control, with the sequence 3'-CCTCTTACCTCAGTTA-CAATTTATA-5' targeting a human beta-globin intron. All MOs were purchased from Gene Tools and injected at the single cell stage into AB wildtype (WT) embryos as follows: 4 ng of *brca2* MO, 1 ng of *adar1* MO, 6 ng of control standard MO and 4 ng/1 ng of *brca2/adar1* MOs per embryo. Embryos were checked at epiboly stage and all unfertilized eggs were discarded, only fertilized eggs were raised and analyzed at 48 h post fertilization (hpf). Injections were performed at least three times with more than 30 embryos per group per experiment.

**Phenotype scoring.** Embryos were raised and analyzed at 48 hpf for morphological phenotypes, that were scored under a dissecting scope. Embryos were grouped into four categories: normal, mild, severe and dead as shown in the corresponding figure.

**Acridine orange staining and analysis.** 48 hpf embryos were anesthetized with 0.03% tricaine and incubated in fish water containing 5 µM acridine orange (AO) for 20 min in the dark. After wash, they were embedded in 1.4% low melting point agarose, and imaged with a Leica SP8 confocal microscope. Fluorescent corpses were counted blindly within a length of ≈ 600 µm spanning from the first somite and through the yolk sac extension along the anterior-posterior (AP) axis of the embryo. Image analyses and AO-positive cells counting were performed using Image J software.

## Statistical analyses

No statistical methods were used to predetermine sample size and experiments were not randomized. Biological replicates are indicated as *n* and technical replicates as N. All graphs show either mean or median values with error bars (standard deviation, SD) calculated using GraphPad Prism 10 software; 95% confidence intervals were used and significance was considered when $*P < 0.05$, $**P < 0.01$, $***P < 0.001$, $****P < 0.0001$; ns, not significant.

## Reporting summary

Further information on research design is available in the Nature Portfolio Reporting Summary linked to this article.

# Data availability

**Materials availability**: All new unique reagents generated in this study are available from the corresponding authors with a completed materials transfer agreement.

**Datasets availability:** The RNA-Seq data generated as part of this study have been deposited in the European Genome-Phenome Archive (EGA), under accession number EGAS50000000518. The datasets on EGA will be made available to interested researchers under limited access on a project-specific basis, subject to approval by the Gustave Roussy Data Access Committee in compliance with the data access agreement terms. Requests should be directed to the corresponding authors. Upon establishment of the data transfer agreement, EGA data release can be expected within 3 business days. Once access has been granted, the period for which the data can be downloaded is flexible and will be defined according to the downloader's needs.

The remaining data are available within the article or Supplementary Information. A Source Data file containing the raw data underlying all figure panels, including full uncropped and unprocessed scans of all blots, is provided along with this article. Any additional information required to reanalyze the data reported in this study is available from the corresponding authors upon request. This study

does not report original code. Source data are provided with this paper.

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

## Acknowledgements

We dedicate this work to patients and their families. We particularly thank donors to the Association Ruban Rose and Fondation pour la Recherche Médicale for their financial support, which has been instrumental in the pursuit of this project. We also thank Dr. Charles M Rice and Dr. Robert Samstein for their material contribution. This work was funded by project grants to S.P.V., R.M.C. and P.P. from Institut National du Cancer (INCa, 2023-165-PLBIO23-139), Fondation pour la Recherche sur le Cancer (Fondation ARC, ARCPGA2023010005866_6378), Association Ruban Rose (Prix Ruban Rose Avenir 2023 to R.M.C.) and Fondation pour la Recherche Médicale (FRM, Prix Raymond Rosen 2023 to S.P.V.). This work was also funded by programme grants to S.P.V. from European Research Council (ERC Starting Grant, TargetSWitch, grant no. 101077864), Fondation ARC (PGA1-RF20190208576), INSERM ATIP-Avenir / La Ligue Contre le Cancer 2018, PRT-K20-067 (ARIANES-CTC, funded by INCa and DGOS), and as part as of programme funding to Gustave Roussy from Institut National du Cancer (INCa-DGOS-Inserm_12551 SIRIC2 and INCa-DGOS-Inserm-ITMO Cancer_18002 SIRIC EpiCURE). Some of the work carried out by R.M.C. at The Institute of Cancer Research was funded by a programme grant to C.J.L. from Breast Cancer Now as part of programme funding to the Breast Cancer Now Toby Robins Research Centre.

## Author contributions

Conceptualization and design of the study: R.M.C., C.J.L., S.P-V. Development of methodology: R.M.C., M.A., C.J.L., S.P-V. Acquisition of data: R.M.C., L.S., M.L-T., J.Z., V.K., A.G., M.G., E.E. Analysis and interpretation of data: R.M.C., L.S., M.L-T., M.A., V.K., M.T., C.D., Y-L.L., P.P., C.J.L., S.P.V. Manuscript drafting: R.M.C., C.J.L., S.P-V. Editing and approval of the final submitted manuscript: R.M.C., L.S., M.L-T., M.A., J.Z., V.K., A.G., M.G., E.E., L.B., D.B.K., R.B., S.J.P., T.T-B., R.S., C.M., M.D., A.N.J.T., F.B., Y.L., S.D., M.T., C.D., Y-L.L., B.P., P.P., C.J.L., S.P-V. Study supervision and funding acquisition: R.M.C., P.P., C.J.L., S.P-V.

## Competing interests

R.M.C. makes the following disclosures: is named inventor on a patent describing the use of ADAR1 inhibitors as anti-cancer agents (WO/2024/189433). J.Z. makes the following disclosures: has received research funding and travel subsidies from Menarini. B.P. makes the following disclosures: has received research funding as part of an institutional program from: Astra Zeneca, Daiichi Sankyo, Gilead, Seagen, MSD; receives and/or has received consulting fees from: Astra Zeneca (institutional), Seagen (institutional), Gilead (institutional), Novartis (institutional), Lilly (institutional), MSD (institutional), Pierre Fabre (personal), Daiichi Sankyo (institutional/personal), Olema (institutional); has received travel subsidies from: Astra Zeneca; Pierre Fabre; MSD; Daiichi Sankyo, Pfizer. A.N.J.T. makes the following disclosures: Receipt of grants / research support: AstraZeneca research costs associated with OlympiA, Myriad Genetics research support for TNT Trial, Breast Cancer Now research grant to Institution, CRUK research grant to Institution. Receipt of honoraria or consultation fees: AstraZeneca, Inbiomotion, Gilead, Innovation in Breast Cancer Symposium, SABCS, GBCC, Cancer Panel, Research to Practise, AACR, Aicme, Penn Medicine, PAGE Therapeutics, Ellipses, VHIO, Dana Faber David Livingston Memorial Symposium, Tango Therapuetics, Guardant, CRUK, Neogenomics, Merck. Educational seminar: Guardant. Stock option: InBioMotion. Royalties: Benefits from ICR's Inventors Scheme associated with patents for use of several PARP inhibitors in DNA repair-deficient cancers held by AstraZeneca with royalty payments to A.N.J.T.'s personal account and research accounts at the Institute of Cancer Research, and to the Institute of Cancer Research. Institutional financial interests: Royalties as above from AstraZeneca. Research costs from AstraZeneca, Myriad Genetics. Non-financial interests: Leadership/Guideline Advisor roles: Director of Breast Cancer Now Research Centre—ICR/KCL, St Gallen Early Breast Cancer Guidelines Panel, ESMO Early Breast Cancer Guidelines Committee, ESMO Scientific Committee, Chair—CRUK Clinical Research Committee, BCRF SAB member, Strategy Committee UK NCRI, Trustee The Cridlan Ross Smith Charitable Trust. C.J.L. makes the following disclosures: receives and/or has received research funding from: AstraZeneca, Merck KGaA, Artios, Neophore; has received consultancy, SAB membership or honoraria payments from: FoRx, Syncona, Sun Pharma, Gerson Lehrman Group, Merck KGaA, Vertex, AstraZeneca, Tango Therapeutics, 3rd Rock, Ono Pharma, Artios, Abingworth, Tesselate, Dark Blue Therapeutics, Pontifax, Astex, Neophore, Glaxo Smith Kline, Dawn Bioventures, Blacksmith Medicines, ForEx; has stock in: Tango, Ovibio, Hysplex, Tesselate, Ariceum. C.J.L. is also named inventor on patents describing the use of DNA repair inhibitors and stands to gain from their development and use as part of the ICR "Rewards to Inventors" scheme and also reports benefits from this scheme associated with patents for PARP inhibitors paid into C.J.L.'s personal accounts and research accounts at the Institute of Cancer Research. C.J.L. is also named inventor on a patent describing the use of ADAR1 inhibitors as anti-cancer agents (WO 2024/189433 A1). S.P.V. makes the following disclosures: has received research funding from Hoffman La Roche and AstraZeneca for unrelated research projects. As part of the Drug Development Department (DITEP), S.P.V. is principal investigator or sub-investigator of clinical trials from Abbvie, Agios Pharmaceuticals, Amgen, Argen-X Bvba, Arno Therapeutics, Astex Pharmaceuticals, Astra Zeneca, Aveo, Bayer Healthcare Ag, Bbb Technologies Bv, Blueprint Medicines, Boehringer Ingelheim, Bristol Myers Squibb, Celgene Corporation, Chugai Pharmaceutical Co., Clovis Oncology, Daiichi Sankyo, Debiopharm S.A., Eisai, Eli Lilly, Exelixis, Forma, Gamamabs, Genentech, Inc., GlaxoSmithKline, H3 Biomedicine, Inc, Hoffmann La Roche Ag, Innate Pharma, Iris Servier, Janssen Cilag, Kyowa Kirin Pharm. Dev., Inc., Loxo Oncology, Lytix Biopharma As, Medimmune, Menarini Ricerche, Merck Sharp & Dohme Chibret, Merrimack Pharmaceuticals, Merus, Millennium Pharmaceuticals, Nanobiotix, Nektar Therapeutics, Novartis Pharma, Octimet Oncology Nv, Oncoethix, Onyx Therapeutics, Orion Pharma, Oryzon Genomics, Pfizer, Pharma Mar, Pierre Fabre, Roche, Sanofi Aventis, Taiho Pharma, Tesaro Inc, and Xencor. S.P.V. has participated to advisory boards for Merck KGaA. S.P.V. is also named inventor on a patent describing the use of ADAR1 inhibitors as anti-cancer agents (WO 2024/189433 A1). All other authors have no conflicts of interest or financial interests to disclose.

## Additional information

Roman M. Chabanon [1,2,3,4] ✉, Liudmila Shcherbakova [1,2], Magali Lacroix-Triki [5,6], Marine Aglave [7], Jean Zeghondy [8], Victor Kriaa [9], Antoine Gougé [1,2], Marlène Garrido[1,2], Elodie Edmond[10], Ludovic Bigot[11], Dragomir B. Krastev [3,4], Rachel Brough[3,4], Stephen J. Pettitt [3,4], Thibault Thomas-Bonafos[1,2,12], Robert Samstein [13,14,15], Christophe Massard[12], Marc Deloger [7], Andrew NJ Tutt [4,16], Fabrice Barlesi [8], Yohann Loriot [11,12], Suzette Delaloge [8], Marcel Tawk [9], Cindy Degerny [9], Yea-Lih Lin [17,18], Barbara Pistilli [8], Philippe Pasero [17,18], Christopher J. Lord [3,4] ✉ & Sophie Postel-Vinay [1,2,12,19] ✉

[1]The ERC (Epi)Genetic Vulnerabilities in Solid Tumors and Sarcoma Laboratory, Inserm Unit UMR981, Gustave Roussy, Villejuif, France. [2]Faculté de Médecine, Université Paris-Saclay, Le Kremlin Bicêtre, France. [3]The CRUK Gene Function Laboratory, The Institute of Cancer Research, London, UK. [4]The Breast Cancer Now Toby Robins Breast Cancer Research Centre, London, UK. [5]Molecular Characterization of Breast and Gynecological Cancers Laboratory, Inserm Unit UMR981, Gustave Roussy, Villejuif, France. [6]Department of Pathology, Gustave Roussy, Villejuif, France. [7]Bioinformatics (BiGR) Platform, Gustave Roussy, Villejuif, France. [8]Department of Medical Oncology, Gustave Roussy, Villejuif, France. [9]Inserm Unit U1195, University Paris-Saclay, Le Kremlin Bicêtre, France. [10]Experimental and Translational Pathology (PETRA) Platform, AMMICa Unit (CNRS Unit UMS3655, Inserm Unit US23), Gustave Roussy, Villejuif, France. [11]Adaptive Resistance to Anti-Cancer Therapies Laboratory, Inserm Unit UMR981, Gustave Roussy, Villejuif, France. [12]Drug Development Department (DITEP), Gustave Roussy, Villejuif, France. [13]Department of Radiation Oncology, Memorial Sloan Kettering Cancer Center, New York, USA. [14]Department of Radiation Oncology, Mount Sinai Hospital, New York, USA. [15]Precision Immunology Institute at Icahn School of Medicine at Mount Sinai, New York, USA. [16]The Breast Cancer Now Research Unit, Guy's Hospital Cancer Centre, King's College London, London, UK. [17]The Ligue Contre Le Cancer Maintenance of Genome Integrity during DNA Replication Laboratory, CNRS Unit UMR9002, Institut de Génétique Humaine, Montpellier, France. [18]Université de Montpellier, Montpellier, France. [19]The University College of London Cancer Institute, University College of London, London, UK. ✉e-mail: roman.chabanon@gustaveroussy.fr; chris.lord@icr.ac.uk; sophie.postel-vinay@gustaveroussy.fr

