## [Transparent Peer Review file · Nature Communications]

Autocrine interferon poisoning mediates ADAR1-dependent synthetic lethality in BRCA1/2-mutant cancers

Corresponding Author: Dr Roman Chabanon

Version 0:

Reviewer comments:

Reviewer #1

(Remarks to the Author)

Authors in this manuscript uncover a synthetic lethal (SL) relationship between mutations in BRCA1/2 (mBRCA) and loss of ADAR1 p150. This phenotype has been rigorously validated across multiple cell lines with diverse BRCA mutations and a zebrafish model. Mechanistically authors attribute this phenotype partially to R-loops and primarily to double-strand RNA induced ISG and IFN response mediated cell killing. Overall, the data is well presented. However, this reviewer's major concerns are the lack of appropriate discussion for the observations made and missing conceptual basis for many conclusions that don't seem to be supported by the data presented. These concerns are listed below:

1. Authors attribute the SL between ADAR1 and mBRCA primarily to dsRNA response mediated by LGP2 and PKR and suggest secondary contributions of R-loop. However, expression of both RNaseH1 (Fig. 3N) and loss of c-GAS (4H) (which binds to RNA-DNA hybrid and not dsRNA) completely rescues the SL phenotype. Hence, assigning a secondary contribution to the R-loop is not supported by the author's data.
2. The confusion for point 1 is further exaggerated by the observation that the SL is owing to cytoplasmic/nuclear p150 but not with p110 isoform of ADAR-1. P110 is much more abundant compared to p150 and hence should be sufficient to resolve R-loop in the nucleus. Hence, it's unclear how R-loops are contributing to the SL phenotype as evidenced by the strong rescue in viability upon expression of RH1 or loss of c-GAS. Are there instances where the function of p110 is reliant on the presence of p150 isoform?
3. Beyond deamination ADAR1 also has a role in recruiting helicases to R-loop. It is therefore imperative that the author demonstrate the role of deaminase activity in the SL phenotype using a catalytic dead ADAR mutant.
4. Can authors comment why ADAR is upregulated in mBRCA compared to BRCA-proficient counterparts? And what can be the likely source of dsRNA. Along similar lines, it is critical to examine how treatment with the transcription inhibitor, DRB, impacts the SL phenotype and activation of IFN and ISG response.
5. S9.6 has now been widely established across the literature to be highly nonspecific. It is therefore critical for the authors to quantify R-loop using the D210N RNaseH probe.
6. PKR impacts ISG response. How do authors explain the reduction in p-IRF3 upon PKR loss?

Minor comments:

1. Does the SL hold true in high-grade serous ovarian cancer cells with BRCA mutation. This analysis is important to extend the application of the findings made by authors.
2. DHX9 is known to resolve R-loop. Why does loss of DHX9 not impact mBRCA sensitivity in the screen. Can be owing to redundancy with our helicases but this point should be included in the discussion.
3. Fig. 3K: γ H2AX levels are higher in BRCA reversion cells-this can be owing to the established non-reliability of quantifying γ H2AX using western. Authors have more convincing γ H2AX data using IF. Hence authors should consider

removing the non-reliable yH2Ax blot to avoid confusion to the readers.

Fig4A: Can the leakiness of Dox-shRNA explain the pIRF3 levels in mBRCA cells with no-Dox ADAR1 shRNA? If so, discuss this in results.

Fig.4I: Why is there no increase in pIRF3 levels upon ADAR1 depletion in mBRCA SUM cells.

Line 202: "Interestingly, higher cytoplasmic ADAR1p150 expression was also associated with increased tumor-infiltrating lymphocytes, regardless of the BRCA1 gene status."

Shouldn't higher ADARp150 should result in lower TILs. How do authors explain this correlation?

Reviewer #2

(Remarks to the Author)

In this manuscript, the authors show that loss of ADAR1 is synthetic lethal with BRCA1/2 deficiency in cell lines and zebra fish model. This synthetic lethal effect is convincingly demonstrated. The authors also show that ADAR1 is overexpressed in BRCA1 mutant tumors, and ADAR1-dependent RNA editing activity is increased in BRCA2 mutant tumor models. Both BRCA1 and ADAR1 are known to suppress R-loop accumulation. As expected, the authors show that loss of ADAR1 in BRCA1 mutant cells increases DNA damage in an R-loop dependent manner. Importantly, expression of RNaseH1 rescued the viability of ADAR1-depleted BRCA1 mutant cells, arguing that R-loops contribute to synthetic lethality. ADAR1 loss is also known to activate dsDNA responses, including type-I interferon response and PKR-mediated stress granules and integrated stress response. As expected, these dsRNA responses are enhanced by the depletion of ADAR1 in BRCA1 mutant cells. Inhibition of these dsRNA responses also rescued the viability of ADAR1 depleted BRCA1 mutant cells even when RNaseH1 was expressed, leading the authors to conclude that autocrine interferon responses are primary cause of this synthetic lethality.

Although most of the experiments in this manuscript are technically well done, the model from this study has a conceptual gap. Because both BRCA1 and ADAR1 are known to suppress R-loops, it is expected that depletion of ARAD1 in BRCA1 mutant cells increases R-loops to induce DNA damage and synthetic lethality. The data in Fig. 3 are overall convincing (but not particularly novel). However, it is surprising that the dsRNA responses are also required for the synthetic lethality because ADAR1 depletion should increase dsRNA in both BRCA1 mutant and WT/Rev cells. Why BRCA1 mutant cells have stronger dsRNA responses after ADAR1 depletion is not explained. This is a major gap in the model that needs to be filled.

Specific comments:

1. As discussed above, the data in Fig. 4A-4F are surprising. Why are BRCA1 mutant cells more responsive to dsRNA? The observations in these panels are not explained.
2. In Fig. 4G and H, it is surprising that depleting RIG1, MDA5, LGP2, and PKR individually reversed the ADAR1-BRCA1 synthetic lethality similarly. Should these proteins respond to dsRNA through independent pathways? If so, one would expect some redundancy among these proteins. It is surprising that depletion of all these proteins individually can suppress the synthetic lethality substantially.
3. In Fig. 3M and N, RNaseH1 rescued the viability of siADAR1-BRCA1 mutant cells almost completely. From the extents of rescue, the contribution of R-loops to synthetic lethality is similar to those of dsRNA sensors. This is not consistent with the conclusion that dsRNA-induced autocrine interferon responses are the primary cause of synthetic lethality.
4. The data in Fig. S10A is not consistent with Fig. 3N. The rescue of cell survival by RNaseH1 expression was nearly 100% in Fig. 3N, but only ~50% in Fig. S10A. The partial rescue in S10A by RNaseH1 is the reason why JAK1/2i can exert additional rescuing effects.
5. The data in Fig. S10B-D only show that the dsRNA responding pathways are still functional in the presence of RNaseH1. However, they don't necessarily mean that these dsRNA pathways are the primary cause of synthetic lethality.

Reviewer #3

(Remarks to the Author)

The manuscript titled "Autocrine Interferon Poisoning Mediates ADAR1-Dependent Synthetic Lethality in BRCA1/2-Mutant Cancers" by Chabanon et al. presents novel and important findings. The study demonstrates that ADAR1, an RNA editing enzyme, and BRCA1/2 exhibit synthetic lethality. The authors explore the mechanisms underlying this relationship and find that the loss of ADAR1 causes DNA damage due to defective R-loop resolution. However, the primary mechanism involves autocrine interferon poisoning, triggered by the activation of RNA sensors (LGP2 and PKR) in BRCA1/2-mutant tumor cells.

This paper has significant implications for cancer therapy, as it highlights interferon signaling as a key vulnerability and provides a preclinical rationale for targeting ADAR1 as a therapeutic strategy in BRCA1/2-mutant cancers. The authors utilize an impressive variety of systems and animal models to demonstrate their findings. I have only a few minor comments:

1. Previous studies have shown that RNA editing is indeed elevated in BRCA cancers, and this finding can complement the narrative of this paper (see Fumagalli et al. and Paz-Yaacov et al., both from 2015).
2. It is well established that specific coding sites, such as AZIN, which is a target of ADAR1P150, are critical for cancer progression under certain conditions (see for example PMID: 23291631). The authors should discuss the possibility that, in addition to double-stranded RNA, a lack of editing at this target may contribute to the presented phenotype.
3. In Figure #2, the authors present the editing levels. To demonstrate the background levels as well, it would be beneficial to include the editing levels of other mismatches (e.g., C-to-T).

Version 1:

Reviewer comments:

Reviewer #1

(Remarks to the Author)

The authors have satisfactorily addressed all the key concerns. I look forward to seeing this exciting work in-print.

Reviewer #2

(Remarks to the Author)

The authors have a good job addressing most of my comments. However, I am still confused about how R-loops activate autocrine interferon poisoning. Although RNaseH1 reversed the synthetic lethality effectively, several of the PRRs involved are not known sensors of R-loops. Even in the revised model figure, it is not clear how R-loop burden activates the whole group of PRRs. The Western blots in the new Fig. 4I makes the interpretation more difficult. If siRNAs targeting individual PRRs actually reduced multiple PRRs simultaneously, it would be hard to pinpoint which PRR is actually required. Also, if the PKR mediated effects are only driven by dsRNA, the complete reversal of SL by RNaseH1 is surprising (Fig. 3N). The RNaseH1 rescue data in Fig. 3N and S10C are not very consistent. The reversal of SL by RNaseH1 becomes partial in Fig. S10C. Overall this is a very nice study, but some gaps remain noticeable.

Reviewer #3

(Remarks to the Author)

The authors have addressed all of my concerns and have undertaken much of the necessary work.

made.

REVIEWER COMMENTS

Reviewer #1 (Remarks to the Author): with expertise in BRCA-mutant cancer

Authors in this manuscript uncover a synthetic lethal (SL) relationship between mutations in BRCA1/2 (mBRCA) and loss of ADAR1 p150. This phenotype has been rigorously validated across multiple cell lines with diverse BRCA mutations and a zebrafish model. Mechanistically authors attribute this phenotype partially to R-loops and primarily to double-strand RNA induced ISG and IFN response mediated cell killing. Overall, the data is well presented. However, this reviewer's major concerns are the lack of appropriate discussion for the observations made and missing conceptual basis for many conclusions that don't seem to be supported by the data presented. These concerns are listed below:

1. Authors attribute the SL between ADAR1 and mBRCA primarily to dsRNA response mediated by LGP2 and PKR and suggest secondary contributions of R-loop. However, expression of both RNaseH1 (Fig. 3N) and loss of c-GAS (4H) (which binds to RNA-DNA hybrid and not dsRNA) completely rescues the SL phenotype. Hence, assigning a secondary contribution to the R-loop is not supported by the author's data.

Our response: We agree with the reviewer: our data indicate that the key mechanistic element of this synthetic lethality (SL) is the increase in R-loops caused by ADAR1 suppression, as shown by a reversal of the SL by RNase H1 expression. The role of cytosolic pattern recognition receptors (PRR) in the SL is a downstream consequence of the increase in R-loops. Reflecting this, we have revised the Abstract as follows:

Abstract, page 2:

ADAR1 is an RNA editing enzyme which prevents autoimmunity by blocking interferon responses triggered by cytosolic RNA sensors, and is a potential target in immuno-oncology. However, predictive biomarkers for ADAR1 inhibition are lacking. Using multiple in vitro and in vivo systems, we show that BRCA1/2 and ADAR1 are synthetic lethal, and that ADAR1 activity is upregulated in BRCA1/2-mutant cancers. ADAR1 depletion in BRCA1-mutant cells caused an increase in R-loops and consequently, an upregulation of cytosolic nucleic acid sensing pattern recognition receptors (PRR), events which were associated with a tumor cell-autonomous type I interferon and integrated stress response. This ultimately caused autocrine interferon poisoning. Consistent with the key role of R-loops in this process, exogenous RNase H1 expression reversed the synthetic lethality. Pharmacological suppression of cell-autonomous interferon responses or transcriptional silencing of cytosolic nucleic acid sensing PRR were also sufficient to abrogate ADAR1 dependency in BRCA1-mutant cells, in line with autocrine interferon poisoning playing a central part in this synthetic lethality. Our findings provide a preclinical rationale for assessing ADAR1-targeting agents in BRCA1/2-mutant cancers, and introduces a conceptually novel approach to synthetic lethal treatments, which exploits tumor cell-intrinsic cytosolic immunity as a targetable vulnerability of cancer cells.

We think the activation of cytosolic nucleic acid sensing PRR and the subsequent interferon response contribute to the SL but operate downstream of the increase in R-loops. Our data that show that the SL is reversed by the silencing of various PRR (Fig. 4G-I; Suppl. Fig. 8J-M) or exposure to JAK/STAT pathway inhibitors (Fig. 4J, K; Suppl. Fig. 8N; Suppl. Fig. 9A-D; Suppl. Fig. 10) does not necessarily mean that the PRR aspect of the SL operates independently of an increase in R-loops. Indeed, there is prior evidence that increased R-loops activate cytosolic nucleic acid sensing PRR, including:

- (i) Sensing of R-loop-derived cytosolic RNA:DNA hybrids by cGAS (Crossley Nature 2023; Mankan EMBO J 2014), RIG1, MDA5 (Crossley Nature 2023), PKR (Wang Genome Res 2018), TLR3 (Crossley Nature 2023), TLR9 (Rigby EMBO J 2014) or

NLRP3 (Vanaja PNAS 2014), which can individually or collectively cross-activate other PRR pathways (Cao Nat Rev Immunol 2016);

- (ii) Sensing of R-loop-derived cytosolic DNA by cGAS (Schumann J Exp Med 2023; Mackenzie EMBO J 2016; Chatzidoukaki Sci Adv 2021; Maxwell Cell 2024; Zannini Cell Mol Life Sci 2024);
- (iii) Sensing of R-loop-derived cytosolic RNA by MDA5 (Schumann J Exp Med 2023);

We agree that in our first submission, our mechanistic model described in Fig. 4M possibly went beyond what was supported by our data and could have minimized the central role of R-loops. As such, we have substantially revised this model (**New Fig. 4M**, shown below) to display only observations empirically substantiated by our data.

New Fig. 4M. Graphical model summarizing our observations. PRR, pattern recognition receptors.

In light of all of the above, we have also modified the manuscript as follows:

Results section, page 11:

RH1 overexpression in BRCA1-mutant cells reversed their sensitivity to ADAR1 silencing in colony-formation assay (Fig. 3M, N) and to a lesser extent in short-term survival assays (Supplementary Fig. 6A-E).

Results section, page 16:

Since both RH1 overexpression and PRR knockdown reversed the BRCA1/2–ADAR1 SL, we assessed the contribution of R-loops to PRR-driven innate immune responses. To do this, we replicated western blots of dsRNA sensors, type I interferon and ISR markers in SUM149 BRCA1-isogenic cells subjected to (i) ADAR1 silencing and/or (ii) RH1 overexpression and/or (iii) JSPi exposure. Whilst BRCA1-mutant cells were unable to activate a type I interferon response upon ADAR1 silencing when exposed to JSPi (Supplementary Fig. 10A) or co-silencing of RIG1 or LGP2 (Supplementary Fig. 10B), RH1 overexpression also counteracted (albeit partially) the upregulation of PRR, type I interferon and integrated stress response markers elicited upon ADAR1 silencing in BRCA1-mutant cells (Supplementary Fig. 10A, B). Consistent with this, short-term survival assays conducted in the same conditions revealed that RH1 overexpression and exposure to JSPi both contributed to reverse the cytotoxic effects of ADAR1 siRNA in SUM149 BRCA1-Mut cells (Supplementary Fig. 10C).

Altogether, this data supported the idea that increased R-loop burden elicited by ADAR1 suppression associates with PRR-driven cell-autonomous interferon poisoning in BRCAm cells, thereby ultimately causing the BRCA1/2–ADAR1 SL.

Discussion section, page 17:

Mechanistically, we show that the BRCA1/2–ADAR1 SL relies upon R-loop-associated DNA damage (Fig. 3) and the selective activation of an R-loop- and PRR-driven cell-autonomous innate immune response in BRCAm cells (Fig. 4), providing the first evidence for autocrine interferon poisoning as a mechanism of BRCA1/2 SL⁴⁶.

Importantly, these findings delineate a distinctive SL mechanism, which is unlike those of other BRCA1/2-associated SL, including the BRCA1/2–PARPi SL. In line with the recently described immunogenic potential of R-loops^{55–62}, our results support a model (Fig. 4M) in which unresolved R-loops elicited by the loss of ADAR1 function^{4,6,48} activate cytosolic nucleic acid sensing PRR, resulting in a toxic cell-autonomous innate immune response which generates the SL phenotype. ADAR1p150 plays an essential role in this SL and could be responsible for the suppression of nuclear R-loops in BRCAm cells, consistent with its nucleocytoplasmic distribution^{63,64}.

2. The confusion for point 1 is further exaggerated by the observation that the SL is owing to cytoplasmic/nuclear p150 but not with p110 isoform of ADAR-1. P110 is much more abundant compared to p150 and hence should be sufficient to resolve R-loop in the nucleus.

Our response: Whilst we agree that one might intuitively expect the more abundant ADAR1p110 isoform to be responsible for the SL, our data clearly indicate that ADAR1p150 plays an important role. Notably:

- (i) BRCA1/2 silencing confers SL to ADAR1p150-knockout cells (Suppl. Fig. 3I-L),
- (ii) Selective silencing of p150 (i.e. leaving p110 expression intact) confers SL to BRCA1-mutant cells (Suppl. Fig. 3G, H).

To further address this question, we transfected cells with cDNAs encoding either p110 or p150 and found that overexpression of p150, but not p110, reduced the BRCA1–ADAR1 SL (New Suppl. Fig. 3M, N), consistent with a central role for ADAR1p150.

New Suppl. Fig. 3M. Cell survival of SUM149 BRCA1-Mut and BRCA1-Rev cells transfected with ADAR1 siRNA in the context of exogenous overexpression of wildtype ADAR1p110 or ADAR1p150. Cells were transfected with control, non-targeting siRNA (siCTRL), or with an individual siRNA targeting

ADAR1 3'UTR (5 nM), and with plasmid constructs containing no cDNA (empty vector, EV), ADAR1p110 or ADAR1p150 cDNA (100 pg/mL). After transfection, cells were continuously cultured for 7 days, after which cell viability was determined by CellTiter-Glo®. Box-and-whiskers indicate median, lower and upper quartiles, and the min to max range; $n=2$, two-way ANOVA post hoc Šídák's test.

New Suppl. Fig. 3N. Western blot of SUM149 *BRCA1*-Mut and *BRCA1*-Rev cells transfected with *ADAR1* siRNA in the context of exogenous overexpression of wildtype ADAR1p110 or ADAR1p150. Cells were transfected with control, non-targeting siRNA (siCTRL), or with an individual siRNA targeting ADAR1 3'UTR (5 nM), and with plasmid constructs containing no cDNA (empty vector, EV), ADAR1p110 or ADAR1p150 cDNA (100 pg/mL). 72 hours after transfection, cell lysates were generated and western blotted to detect ADAR1 protein. $n=2$. Short and long exposure times were used to image some membranes.

We have now modified the manuscript to include this data:

Results section, page 7:

To further explore the relative contribution of ADAR1 isoforms to the SL, we conducted short-term survival assays in SUM149 BRCA1-isogenic cells subjected to (i) ADAR1 knockdown with an siRNA targeting ADAR1 3'-UTR region and (ii) concomitant exogenous overexpression of ADAR1p110 or ADAR1p150. We found that overexpression of ADAR1p150, but not ADAR1p110, could reverse the BRCA1-ADAR1 SL effects (Supplementary Fig. 3M, N), supporting a key role for ADAR1p150 in this SL.

Hence, it's unclear how R-loops are contributing to the SL phenotype as evidenced by the strong rescue in viability upon expression of RH1 or loss of c-GAS.

Our response: As described earlier, the reversal of the synthetic lethality by RNase H1 overexpression strongly suggests that R-loops are critical to this SL (Fig. 3; Suppl. Fig. 6-7). As ADAR1p150 is both nuclear and cytoplasmic, this isoform could still be responsible for the suppression of nuclear R-loops. With revisions to the main text of the manuscript, we hope to have made this point clear. Our data also indicate a downstream effect of this increased R-loop burden, which is the activation of cytosolic nucleic acid sensing PRR and a resultant type I interferon and integrated stress response (Fig. 4; Suppl. Fig. 8-10). These latter effects contribute to the SL, as altering either PRR-mediated cytosolic nucleic acid sensing (Fig. 4G-I; Suppl. Fig. 8J-M) or downstream interferon signaling (Fig. 4J, K; Suppl. Fig. 8N; Suppl. Fig. 9A-D; Suppl. Fig. 10) can reverse the phenotype. As described above, the idea that an increase in R-loops can activate multiple PRR is substantiated by the work of others.

We have accordingly modified the manuscript to clarify our working model and now highlight the basis of this rationale in the revised Discussion:

Discussion section, page 17:

Importantly, these findings delineate a distinctive SL mechanism, which is unlike those of other BRCA1/2-associated SL, including the BRCA1/2–PARPi SL. In line with the recently described immunogenic potential of R-loops^{55–62}, our results support a model (**Fig. 4M**) in which unresolved R-loops elicited by the loss of ADAR1 function^{4,6,48} activate cytosolic nucleic acid sensing PRR, resulting in a toxic cell-autonomous innate immune response which generates the SL phenotype. ADAR1p150 plays an essential role in this SL and could be responsible for the suppression of nuclear R-loops in BRCAm cells, consistent with its nucleocytoplasmic distribution^{63,64}. Still, some elements of the BRCA1/2–ADAR1 SL mechanism remain to be explored.

Discussion section, page 18:

Although our data point to a central role of R-loops in the mechanism of this SL, multiple cytosolic nucleic acid sensing PRR – not restricted to those described as putative RNA:DNA hybrid sensors (i.e. cGAS^{55,68}, TLR3⁵⁵, TLR9⁶⁹ and NLRP3⁷⁰), are activated by ADAR1 silencing in BRCA1/2-mutant cells. Prior work indicates that R-loop-derived cytosolic nucleic acids – including DNA, dsRNA, and RNA:DNA hybrids, can trigger interferon responses via DNA- and dsRNA-sensing PRR^{55–62}, and conversely, that canonical cytosolic DNA and dsRNA sensors can detect RNA:DNA hybrids^{55,71}, suggesting a possible contribution of several cytosolic nucleic acid species to the SL. Besides, others have shown that cytosolic dsRNA arises from R-loop-prone genomic regions, including telomeres (via telomeric repeat-containing RNA, TERRA)^{4,65,72,73}, and micronuclei (via aberrant transcription of chromosomes trapped in micronuclei)^{48,74}. These latter findings could be consistent with our observations that an increased R-loop burden also, in the context of the BRCA1/2–ADAR1 SL, activates dsRNA-sensing PRR.

Are there instances where the function of p110 is reliant on the presence of p150 isoform?

Our response: This is a very interesting point. To the best of our knowledge, whether instances of p110 function being reliant on p150 exist is unknown. Homodimerization (Cho J Biol Chem 2003; Chilibeck J Biol Chem 2006; Valente J Biol Chem 2007; Sun PNAS 2021) and heterodimerization (Cho J Biol Chem 2003; experiments performed in HeLa cells) of both isoforms have been reported. Whilst homodimerization has been described as being essential for ADAR1 adenosine deaminase activity, the role of heterodimerization is still unclear (Cho J Biol Chem 2003).

To best address the reviewer’s comment, we have performed co-immunoprecipitation of ADAR1 isoforms in the SUM149 model (**Rebuttal Fig. R1**) and U2OS cells (**Rebuttal Fig. R2**), which revealed the presence of p110/p150 heterodimers, suggesting a potential interaction. Further assessing whether the function of p110 relies on p150 was unfortunately not technically feasible within the timeframe allowed for this rebuttal and goes beyond the scope of this manuscript.

Rebuttal Fig. R1. Heterodimerization of ADAR1 isoforms in SUM149 BRCA1-isogenic cells. A. Western blot of ADAR1 on total cell extracts following immunoprecipitation of ADAR1p110/p150 (using

ADAR1 antibody #sc-73408, Santa Cruz) in SUM149 *BRCA1*-Mut and SUM149 *BRCA1*-Rev cells. Input, immunoprecipitated (IP) and flow-through (FT) fractions are shown. **B.** Western blot of ADAR1 on total cell extracts following immunoprecipitation of ADAR1p150 (using ADAR1p150-specific antibody #A303-883A, Bethyl Laboratories) in SUM149 *BRCA1*-Mut and SUM149 *BRCA1*-Rev cells. Input, immunoprecipitated (IP) and flow-through (FT) fractions are shown.

Rebuttal Fig. R2. Heterodimerization of ADAR1 isoforms in U2OS cells. Western blot of ADAR1 on total cell extracts following immunoprecipitation of ADAR1p110/p150 (using ADAR1 antibody #sc-73408, Santa Cruz) or ADAR1p150 (using ADAR1p150-specific antibody #A303-883A, Bethyl Laboratories) in U2OS cells. Input, immunoprecipitated (IP) and flow-through (FT) fractions are shown.

3. Beyond deamination ADAR1 also has a role in recruiting helicases to R-loop. It is therefore imperative that the author demonstrate the role of deaminase activity in the SL phenotype using a catalytic dead ADAR mutant.

Our response: To address this, we have transfected plasmid constructs encoding wildtype (WT), catalytically-inactive (Δ CD), or Z-DNA binding-defective (Δ Z α) ADAR1p150 mutants in SUM149 models, and evaluated the SL phenotype. We found that expression of p150- Δ Z α , but not p150- Δ CD, efficiently reversed the SL phenotype (**New Suppl. Fig. 30**), suggesting that ADAR1p150 deaminase activity is required for the survival of *BRCA1*-mutant cells.

New Suppl. Fig. 30. Cell survival of SUM149 *BRCA1*-Mut and *BRCA1*-Rev cells transfected with *ADAR1* siRNA in the context of exogenous overexpression of wildtype and mutant ADAR1p150. Cells were transfected with control, non-targeting siRNA (siCTRL), or with an individual siRNA targeting

ADAR1 3'UTR (5 nM), and with plasmid constructs containing no cDNA (empty vector, EV), wildtype, catalytically-inactive (Δ CD) or $Z\alpha$ domain-mutated (Δ Z α) ADAR1p150 cDNA (100 pg/mL). After transfection, cells were continuously cultured for 7 days, after which cell viability was determined by CellTiter-Glo®. Box-and-whiskers indicate median, lower and upper quartiles, and the min to max range; n=2, two-way ANOVA post hoc Šidák's test.

This new data has been incorporated in the manuscript, which now reads:

Results section, page 7:

*To next investigate whether ADAR1p150 deaminase activity was involved in the SL, we transfected plasmids encoding wildtype ADAR1p150, the deaminase-defective mutant G1007R (p150- Δ CD), or the Z-DNA binding-defective mutant P193A (p150- Δ Z α)³¹ in SUM149 models. We observed that ADAR1p150- Δ Z α , but not ADAR1p150- Δ CD, significantly reversed the SL effects (**Supplementary Fig. 3O**), indicating the essentiality of ADAR1p150 deaminase activity in BRCA1-mutant cells survival.*

4. Can authors comment why ADAR is upregulated in mBRCA compared to BRCA-proficient counterparts?

Our response: Higher expression of ADAR1 in BRCA1/2-mutant tumors may be explained by the dependency of BRCA1/2-mutant cells upon ADAR1. One reasonable hypothesis is that ADAR1 overexpression compensates for impaired BRCA1/2-mediated repair of R-loop-derived DNA damage, thereby enabling the maintenance of cell homeostasis and survival (Hatchi Mol Cell 2015; Shivji Cell Rep 2018; D'Alessandro Nat Commun 2018). This is consistent with the notion of hyperexpression of BRCA1/2 synthetic lethal partners being a general feature of homologous recombination-defective cancers (Haider Nat Genet 2025).

We have now modified the manuscript to clarify this point:

Results section, pages 8-9:

*First, we built a cohort of 63 patients with TNBC (including 32 BRCA1-wildtype and 31 BRCA1-mutant tumor samples) and optimized an immunohistochemistry assay to evaluate nuclear vs. cytoplasmic ADAR1p150 expression in tumor cells. This revealed that BRCA1-mutant tumors displayed a higher cytoplasmic ADAR1p150 expression (**Fig. 2A, B**; $P = 0.0359$, Mann-Whitney U test), while no significant difference was observed regarding ADAR1p150 nuclear expression (**Supplementary Fig. 4A**; $P = 0.3976$, not significant). Interestingly, higher cytoplasmic ADAR1p150 expression was also associated with increased tumor-infiltrating lymphocytes, regardless of the BRCA1 gene status (**Fig. 2C**; **Supplementary Fig. 4B-E**). As the upregulation of BRCA1/2 synthetic lethal partners is a general feature of HR-defective cancers⁴⁰, one reasonable assumption is that this higher ADAR1 expression seen in BRCA1-mutant cancers reflects the dependency these cancers have upon ADAR1 function.*

And what can be the likely source of dsRNA.

Our response: We do not show nor conclude that dsRNA levels are increased when ADAR1 is silenced. Our data indicate that multiple cytosolic nucleic sensing PRR are activated by ADAR1 silencing in BRCA1/2-mutant cells, most likely as a result of the increase in R-loops. Whilst some of these PRR are known to detect RNA:DNA hybrids (e.g. cGAS, Crossley Nature 2023; RIG1 or PKR, Wang Genome Res 2018), others such as the canonical dsRNA sensors LGP2 or MDA5, do not. We currently cannot precisely explain how, in the context of the BRCA1/2-ADAR1 SL, these latter PRR are activated as a consequence of R-loops. But as described above, the idea that an increase in R-loops causes cytosolic nucleic acids accumulation, leading to the activation of dsRNA-sensing PRR has precedence in the literature (please see our response to the reviewers' point 1).

Because we do not empirically show that dsRNA is part of the mechanism underlying the *BRCA1/2*–*ADAR1* SL, we have restricted our mechanistic model to reflect only what is substantiated by our data (**New Fig. 4M**).

New Fig. 4M. Graphical model summarizing our observations. PRR, pattern recognition receptors.

We have also accordingly expanded the manuscript Discussion to better highlight our model's limitations:

Discussion section, page 18:

Related to this, the exact nature of cytosolic nucleic acid species triggering cell-autonomous interferon poisoning remains to be defined. Although our data point to a central role of R-loops in the mechanism of this SL, multiple cytosolic nucleic sensing PRR – not restricted to those described as putative RNA:DNA hybrid sensors (i.e. cGAS^{55,68}, TLR3⁵⁵, TLR9⁶⁹ and NLRP3⁷⁰), are activated by ADAR1 silencing in BRCA1/2-mutant cells. Prior work indicates that R-loop-derived cytosolic nucleic acids – including DNA, dsRNA, and RNA:DNA hybrids, can trigger interferon responses via DNA- and dsRNA-sensing PRR^{55–62}, and conversely, that canonical cytosolic DNA and dsRNA sensors can detect RNA:DNA hybrids^{55,71}, suggesting a possible contribution of several cytosolic nucleic acid species to the SL. Besides, others have shown that cytosolic dsRNA arises from R-loop-prone genomic regions, including telomeres (via telomeric repeat-containing RNA, TERRA)^{4,65,72,73}, and micronuclei (via aberrant transcription of chromosomes trapped in micronuclei)^{48,74}. These latter findings could be consistent with our observations that an increased R-loop burden also, in the context of the BRCA1/2–ADAR1 SL, activates dsRNA-sensing PRR.

Along similar lines, it is critical to examine how treatment with the transcription inhibitor, DRB, impacts the SL phenotype and activation of IFN and ISG response.

Our response: We have now assessed the impact on SL phenotypes and downstream activation of interferon responses of the RNAPII inhibitor DRB, used at non-toxic concentrations compatible with an inhibition of transcription elongation (Zhang Nat Commun 2022). We found that exposure to DRB did neither affect the SL effects (**Rebuttal Fig. R3**) nor the activation of interferon responses (**Rebuttal Fig. R4**) elicited by ADAR1 silencing in *BRCA1*-mutant cells, though DRB itself appeared to modulate the basal expression of certain PRR. This might be related to the mechanism of action of DRB which inhibits transcription by blocking the release of RNAPII into elongation, thereby enforcing RNAPII promoter-proximal pausing. The latter causes (i) an increase in paused-associated R-loops at promoters and (ii) a decrease in elongation-associated R-loops along gene bodies (Castillo-Guzman DNA Repair

2021). These dual effects of DRB towards R-loop formation at different genomic sites thereby complexifies the interpretation of its effects on the *BRCA1/2-ADAR1* SL phenotypes.

Rebuttal Fig. R3. Effect of DRB on the *BRCA1-ADAR1* synthetic lethality in the SUM149 model. Cell survival of SUM149 *BRCA1*-Mut cells transfected with *ADAR1* siRNA in the context of exposure to the RNA polymerase II inhibitor (RNAPII) DRB. Cells were either transfected with control, non-targeting siRNA (siCTRL) or with a titration of *ADAR1* siRNA SMARTpool (nM). 24 hours after transfection, cells were exposed to mock control (DMSO) or DRB (10 μ M) and continuously cultured for 7 days, after which cell viability was determined by CellTiter-Glo®. PLK1-targeting siRNA (siPLK1) was used as a positive control. Box-and-whiskers indicate median, lower and upper quartiles, and the min to max range; $n=2$, two-way ANOVA post hoc Dunnett's test.

Rebuttal Fig. R4. Effect of DRB on the cytosolic nucleic acid sensing / type I interferon responses in the SUM149 model. Western blot of SUM149 *BRCA1*-Mut cells subjected to transfection of *ADAR1* siRNA in the context of exposure to the RNA polymerase II inhibitor (RNAPII) DRB. Cells were either transfected with control, non-targeting siRNA (siCTRL) or with a titration of *ADAR1* siRNA SMARTpool (nM). 24 hours after transfection, cells were exposed to mock control (DMSO) or DRB (10 μ M). 72 hours

after transfection, cell lysates were generated and western blotted to detect RIG1, MDA5, LGP2, phosphorylated PKR (p-PKR), PKR, cGAS, phosphorylated IRF3 (p-IRF3), IRF3, phosphorylated STAT1 (p-STAT1) and STAT1 proteins. $n=2$.

5. S9.6 has now been widely established across the literature to be highly nonspecific. It is therefore critical for the authors to quantify R-loop using the D210N RNaseH probe.

Our response: We agree with the reviewer that the S9.6 antibody has been reported to have limited specificity but chose to use that antibody since it has previously been repeatedly employed in the literature to show that ADAR1 loss causes R-loop accumulation (Shiromoto Nat Commun 2021; Tang Nature 2023; Zhang Nucleic Acids Res 2023).

To best address the reviewer's comment, we attempted to generate SUM149 cells stably expressing catalytically-inactive RNase H1 (D210N mutated, V5-tagged RNase H1 for use as a probe in immunofluorescence assays; Chen Mol Cell 2017). Clonal populations could unfortunately not be successfully obtained during the timeframe allowed for this rebuttal. We therefore chose to use RNase H1 overexpression to functionally explore the role of R-loops. Still, the reviewer raises a very valid point and this could be included in a further, more mechanistically detailed, piece of work.

We now acknowledge this limitation in the revised manuscript Discussion:

Discussion section, pages 18-19:

Secondly, the characterization of R-loop as the trigger of ADAR1-dependent SL effects would require further investigation to determine the nature (promoter-paused vs. elongation-associated R-loops⁷⁵), genomic location (promoter proximal, exonic or non-coding regions) and functional context (co-transcriptional vs. DNA repair-associated R-loops at DSBs⁷⁶⁻⁷⁸ or replication forks⁷⁹) of R-loops involved in the BRCA1/2-ADAR1 SL. The use of orthogonal methods to visualize (e.g. catalytically-inactive mutant RNase H1 protein) and capture both native (e.g. R-loop CUT&Tag) and ex cellulo-isolated R-loops (e.g. DRIP-Seq) would allow a more comprehensive profiling of the R-loop landscape in response to impaired ADAR1 function.

6. PKR impacts ISG response. How do authors explain the reduction in p-IRF3 upon PKR loss?

Our response: We have now re-evaluated this phenotype in new experiments (**New Fig. 4I**) and found that PKR silencing did not reduce IRF3 phosphorylation levels to the same extent as in our initial observations. This variability in results may be explained by the fact that PKR is an interferon-stimulated gene (Sadler Nat Rev Immunol 2008), which also amplifies type I interferon production following viral infection (McAllister J Biol Chem 2012; Schulz Cell Host Microbe 2010). This feedback regulation makes the interpretation of the modulation of pIRF3 upon PKR loss more complex.

New Fig. 4I. Western blot of SUM149 *BRCA1*-Mut and *BRCA1*-Rev cells subjected to co-transfection with *ADAR1* siRNA and one of a series of siRNAs targeting RNA and RNA:DNA hybrid sensors. $n=3$.

We apologize for this and have now replaced the original data with **New Fig. 4I**, as well as corrected the manuscript accordingly:

Results section, page 14:

Consistent with this, we observed that co-transfection of LGP2 siRNA was sufficient to abrogate ADAR1 siRNA-induced phosphorylation of PKR and IRF3, while co-transfection of RIG1 or MDA5 siRNA only partially hindered these effects (Fig. 4I). Co-transfection of PKR siRNA did not affect ADAR1 siRNA-induced phosphorylation of IRF3, in line with the notion that PKR is activated by and acts downstream of type I interferon signaling⁵⁰.

Minor comments:

1. Does the SL hold true in high-grade serous ovarian cancer cells with BRCA mutation. This analysis is important to extend the application of the findings made by authors.

Our response: We acknowledge the importance and clinical relevance of assessing the *BRCA1/2-ADAR1* SL in other *BRCA1/2*-associated malignancies, such as high-grade serous ovarian cancer. To do so, we evaluated the cytotoxic effects of silencing *ADAR1* in the previously-described ID8 *Brca1*-isogenic murine ovarian carcinoma model (Walton Sci Rep 2017; **New Fig. 1H**). We found that siRNA targeting of *ADAR1* conferred SL effects to *Brca1*-mutant ID8 cells (**New Suppl. Fig. 2K-M**), which were more modest than those observed in TNBC models (**New Fig. 1Q**).

Still, our revalidation of the *BRCA1/2-ADAR1* SL in 14 independent models across multiple histotypes and species (**New Fig. 1H**, **New Fig. 1Q**) supports that these effects are, at least in part, independent of molecular background or tissue-of-origin – though it remains possible that the SL may be more profound in TNBC, which is the current main focus of our manuscript.

New Suppl. Fig. 2K. Dose-response survival curves of ID8 *Brca1*-wildtype (WT) and *Brca1*-knockout (KO) cells exposed to increasing concentrations of the PARPi talazoparib for 7 days. Mean \pm SD; N=4. **L.** Cell survival of ID8 *Brca1*-wildtype (WT) and *Brca1*-knockout (KO) cells transfected with *Adar1* siRNA. Cells were transfected with control, non-targeting siRNA (siCTRL), or with a titration of *Adar1* siRNA SMARTpool (nM). After transfection, cells were continuously cultured for 7 days, after which cell viability was determined by CellTiter-Glo®. PLK1-targeting siRNA (siPLK1) was used as a positive control. Box-and-whiskers indicate median, lower and upper quartiles, and the min to max range; $n=2$, two-way ANOVA post hoc Dunnett's test. **M.** Western blot of ID8 *Brca1*-wildtype (WT) and *Brca1*-knockout (KO) cells transfected with *Adar1* siRNA as described in L. 72 hours after transfection, cell lysates were generated and western blotted to detect ADAR1 protein. $n=2$.

New Figure 1H. Schematic describing the isogenic and non-isogenic cell line models used throughout the study. **Q.** Heatmap showing the relative sensitivity to ADAR1 silencing of all models evaluated in the study, based on the presented survival assays. Color mapping indicates the corresponding surviving fractions (SF) calculated based on results from Fig. 1 and Supplementary Fig. 1-3 (blue, SF > 0.8; red, SF < 0.8). The species (Hs, human; Mm, mouse; Dr, zebrafish), histology (TNBC, triple-negative breast cancer; Kidney, embryonic kidney; Retina, normal retina; CRC, colorectal cancer; Fibro, embryonic fibroblast; BC, breast cancer; OC, ovarian cancer) and type of assay (RNAi; CRISPRn; MO) are indicated to the left. WT, wildtype; Rev, revertant; KO, knockout; Mut, mutant.

This new data has been incorporated in the manuscript, which now reads:

Results section, pages 6-7:

To explore whether the BRCA1–ADAR1 SL effect was private to SUM149 cells or more penetrant, we assessed the effects of siRNA- or CRISPR-Cas9-mediated targeting of ADAR1 in ten additional independent models (**Fig. 1H**): (i) four isogenic systems of BRCA1 deficiency, including mouse (*Mus musculus*, Mm) embryonic fibroblasts (MEFs)²³, human retinal pigment epithelial cells (RPE1)²⁴, mouse mammary carcinoma cells (4T1)²⁵ or mouse ovarian carcinoma cells (ID8)²⁶; and (ii) a molecularly diverse, non-isogenic panel of six human TNBC cell lines with/without endogenous BRCA1 mutations (BRCA1-wildtype: MDA-MB-231, CAL51, CAL120, Hs578T; BRCA1-mutant: MDA-MB-436, HCC1937). The homologous recombination (HR) status of these cell lines was confirmed by assessing PARPi sensitivity (**Supplementary Fig. 1-2**). In all models, BRCA1-mutant cells were significantly more sensitive to ADAR1 silencing than BRCA1-wildtype cells (**Fig. 1I, J**; **Supplementary Fig. 1N**; **Supplementary Fig. 2**). In assessing whether ADAR1 SL effects extended to BRCA2-mutant cells, we found that ADAR1 silencing elicited SL in two different BRCA2-isogenic systems: human BRCA2-knockout colorectal carcinoma cells (DLD1)²⁷ and mouse *Brca2*-knockout 4T1 cells²⁵ (**Supplementary Fig. 2G-J, 3A-F**). We also confirmed the BRCA1/2–ADAR1 SL effect in isogenic ADAR1-wildtype or -knockout human embryonic kidney cells (HEK293T) subjected to BRCA1 or BRCA2 silencing²⁸ (**Fig. 1K, L**).

2. DHX9 is known to resolve R-loop. Why does loss of DHX9 not impact mBRCA sensitivity in the screen. Can be owing to redundancy with other helicases but this point should be included in the discussion.

Our response: Due to the probability of false negatives in screening experiments, the fact that *DHX9* silencing did not impact cell viability in our original screen is not sufficient to evidence the lack of essentiality of this gene in SUM149 cells. To best address this, we conducted an independent clonogenic survival assay evaluating the effect of *DHX9* knockdown on the viability of SUM149 *BRCA1*-isogenic cells (**Rebuttal Fig. R5**). This revealed that cytotoxic effects of *DHX9* silencing operate in both *BRCA1*-mutant and -revertant cells, which is consistent with recent literature reports (Murayama *Cancer Discov* 2024). As highlighted by the reviewer, this lack of selectivity could be owing to redundancy with other helicases or PRR, and we now have included this point in the discussion.

Rebuttal Fig. R5. Cytotoxic effects of DHX9 silencing in SUM149 BRCA1-isogenic cells. A, B. Clonogenic survival of SUM149 *BRCA1*-Mut and *BRCA1*-Rev cells transfected with *DHX9* siRNA. Cells were transfected with control, non-targeting siRNA (siCTRL), or with *DHX9* siRNA SMARTpool (7,5 nM). After transfection, cells were continuously cultured for 14 days, after which colonies were stained and counted. Box-and-whiskers indicate median, lower and upper quartiles, and the min to max range; $n=2$, two-way ANOVA post hoc Dunnett's test.

Discussion section, pages 17-18:

Intriguingly, silencing of *DHX9* (a PRR which both suppresses cytosolic dsRNA sensing^{59,66} and bears a nuclear RNA helicase activity against R-loops⁶⁷) did not elicit *BRCA1*-dependent SL effects in our original screen, suggesting a possible redundancy of *DHX9* function (i) at the nuclear level, with that of other helicases for the resolution of R-loops and/or (ii) at the cytosolic level, with that of other PRR for the suppression of dsRNA sensing-mediated responses.

3. Fig. 3K: γ H2AX levels are higher in *BRCA* reversion cells-this can be owing to the established non-reliability of quantifying γ h2ax using western. Authors have more convincing γ h2ax data using IF. Hence authors should consider removing the non-reliable γ H2Ax blot to avoid confusion to the readers.

Our response: We agree with the reviewer and have now removed γ -H2AX blots from the corresponding figure panels (see **New Fig. 3K, L** and **New Suppl. Fig. 5K, L** below).

New Fig. 3K, L. Western blot of SUM149 *BRCA1*-Mut and *BRCA1*-Rev cells (K) or MEF *Brca1*-wildtype (WT) and *Brca1*-mutant (Δ 11) cells (L) transfected with a concentration range (nM) of *ADAR1* siRNA. $n=2$.

New Suppl. Fig. 5K. Western blot of SUM149 *BRCA1*-Mut and *BRCA1*-Rev cells transduced with a doxycycline-inducible *ADAR1*-targeting shRNA. Cells were either mock-transduced, or transduced with an *ADAR1*-targeting shRNA and subsequently exposed to a titration of doxycycline (ng/mL) for 72 hours. After this, cell lysates were generated and western blotted to detect phosphorylated CHK1 (p-CHK1), CHK1 and cleaved-PARP1 (c-PARP1) proteins. $n=2$. Appropriate silencing of *ADAR1* was verified as shown in Fig. 4A; for the sake of clarity, the *ADAR1* and actin blots were duplicated from Fig. 4A. **L.** Western blot of 4T1 *Brca*-wildtype (WT), *Brca1*- and *Brca2*-knockout (KO) cells transfected with *Adar1* siRNA as described in F-H. 72 hours after transfection, cell lysates were generated and western blotted to detect phosphorylated CHK1 (p-CHK1), CHK1 and cleaved-PARP1 (c-PARP1) proteins. $n=2$.

The text has been modified to reflect those changes:

Results section, page 11:

ADAR1 silencing also caused micronucleation in *BRCAM* but not in *BRCA1-revertant* or *BRCA1/2-wildtype* cells (**Fig. 3E-H; Supplementary Fig. 5I**), suggesting a selective induction of chromosomal instability in *BRCAM* cells. We further noted that *ADAR1* silencing in *BRCA1-mutant* cells caused a significant accumulation of RPA foci in S-phase population (**Fig. 3I, J; Supplementary Fig. 5J**), increased *CHK1* phosphorylation and *PARP1* cleavage as assessed by western blot (**Fig. 3K, L; Supplementary Fig. 5K, L**), while *BRCA1-revertant* cells showed little or no change in these marks, suggesting increased replication stress and apoptosis in *BRCA1-mutant* cells subjected to *ADAR1* knockdown.

Fig4A: Can the leakiness of Dox-shRNA explain the pIRF3 levels in mBRCA cells with no-Dox ADAR1 shRNA? If so, discuss this in results.

Our response: This is a valid point. The leakiness of the system could indeed explain increased IRF3 phosphorylation levels in SUM149 *BRCA1-Mut* sh*ADAR1* in the absence of Dox. We have added a note on this in the revised Results section, which now reads:

Results section, page 13:

We noted increased IRF3 phosphorylation levels of SUM149 *BRCA1-Mut* in response to *ADAR1* shRNA expression, even in absence of doxycycline (**Fig. 4A**), suggesting potential promoter leakage of the shRNA in that cell line.

Fig.4I: Why is there no increase in pIRF3 levels upon ADAR1 depletion in mBRCA SUM cells.

Our response: We agree with the reviewer that the extent of IRF3 phosphorylation observed in Fig. 4I was more modest compared with other figure panels (e.g. Fig. 4A, Suppl. Fig. 8H, Suppl. Fig. 10A). To confirm this phenotype and consolidate the data presented in this manuscript, we have conducted additional independent biological replicates. This new data validates our initial observation i.e. increased IRF3 phosphorylation levels upon *ADAR1* knockdown in SUM149 *BRCA1-Mut* cells (**New Fig. 4I**), and now replaces the original Fig. 4I.

New Fig. 4I. Western blot of SUM149 *BRCA1-Mut* and *BRCA1-Rev* cells subjected to co-transfection with *ADAR1* siRNA and one of a series of siRNAs targeting RNA and RNA:DNA hybrid sensors. *n*=3.

Line 202: "Interestingly, higher cytoplasmic ADAR1p150 expression was also associated with increased tumor-infiltrating lymphocytes, regardless of the BRCA1 gene status." Shouldn't higher ADARp150 should result in lower TILs. How do authors explain this correlation?

Our response: Our observation that increased levels of TILs correlate with increased tumor ADAR1p150 expression is consistent with the fact that p150 is interferon-inducible i.e. increased TILs cause increased intra-tumor production of interferon- γ which in turn results in increased ADAR1p150 expression. This correlative observation suggests that tumor ADAR1p150 expression associates with a more immunogenic tumor microenvironment.

End of Referee #1 comments

Reviewer #2 (Remarks to the Author): with expertise in BRCA-mutant cancer, synthetic lethality

In this manuscript, the authors show that loss of ADAR1 is synthetic lethal with BRCA1/2 deficiency in cell lines and zebra fish model. This synthetic lethal effect is convincingly demonstrated. The authors also show that ADAR1 is overexpressed in BRCA1 mutant tumors, and ADAR1-dependent RNA editing activity is increased in BRCA2 mutant tumor models. Both BRCA1 and ADAR1 are known to suppress R-loop accumulation. As expected, the authors show that loss of ADAR1 in BRCA1 mutant cells increases DNA damage in an R-loop dependent manner. Importantly, expression of RNaseH1 rescued the viability of ADAR1-depleted BRCA1 mutant cells, arguing that R-loops contribute to synthetic lethality. ADAR1 loss is also known to activate dsDNA responses, including type-I interferon response and PKR-mediated stress granules and integrated stress response. As expected, these dsRNA responses are enhanced by the depletion of ADAR1 in BRCA1 mutant cells. Inhibition of these dsRNA responses also rescued the viability of ADAR1 depleted BRCA1 mutant cells even when RNaseH1 was expressed, leading the authors to conclude that autocrine interferon responses are primary cause of this synthetic lethality.

Although most of the experiments in this manuscript are technically well done, the model from this study has a conceptual gap. Because both BRCA1 and ADAR1 are known to suppress R-loops, it is expected that depletion of ADAR1 in BRCA1 mutant cells increases R-loops to induce DNA damage and synthetic lethality. The data in Fig. 3 are overall convincing (but not particularly novel).

Our response: As far as we are aware, our work is the first to report a synthetic lethal interaction between *ADAR1* and *BRCA1/2* and the essential contribution of cell-autonomous interferon poisoning to this synthetic lethality (Fig. 4). To the best of our knowledge, a mechanism of autocrine interferon poisoning has never been reported in any *BRCA1/2*-related, or more generally DNA damage response-related, synthetic lethality.

However, it is surprising that the dsRNA responses are also required for the synthetic lethality because ADAR1 depletion should increase dsRNA in both BRCA1 mutant and WT/Rev cells. Why BRCA1 mutant cells have stronger dsRNA responses after ADAR1 depletion is not explained. This is a major gap in the model that needs to be filled.

Our response: We agree with the reviewer that this is an important point and apologize for not having sufficiently detailed this in our original submission. Our data support that the activation of cytosolic nucleic acid sensing pattern recognition receptors (PRR), which results in interferon poisoning, is key to the synthetic lethality (SL), but that this process is dependent on R-loop accumulation. Consistent with this, RNase H1 overexpression prevents the upregulation of PRR, type I interferon and integrated stress response markers elicited upon ADAR1 silencing in *BRCA1*-mutant cells (Suppl. Fig. 10A, B). Therefore, *BRCA1*-mutant cells – which have an increased R-loop burden following ADAR1 silencing (Fig. 3; Suppl Fig. 5-7), display enhanced R-loop-dependent cytosolic nucleic acid sensing responses compared with *BRCA1*-wildtype cells.

We have now revised the Abstract and graphical model (**New Fig. 4M**) to better explain this functional link, which exposes R-loops and cytosolic nucleic acid sensing responses as key mechanistic nodes underlying the *BRCA1/2*–*ADAR1* SL:

Abstract, page 2:

ADAR1 is an RNA editing enzyme which prevents autoimmunity by blocking interferon responses triggered by cytosolic RNA sensors, and is a potential target in immuno-oncology. However, predictive biomarkers for ADAR1 inhibition are lacking. Using multiple in vitro and in vivo systems, we show that

BRCA1/2 and ADAR1 are synthetic lethal, and that ADAR1 activity is upregulated in BRCA1/2-mutant cancers. ADAR1 depletion in BRCA1-mutant cells caused an increase in R-loops and consequently, an upregulation of cytosolic nucleic acid sensing pattern recognition receptors (PRR), events which were associated with a tumor cell-autonomous type I interferon and integrated stress response. This ultimately caused autocrine interferon poisoning. Consistent with the key role of R-loops in this process, exogenous RNase H1 expression reversed the synthetic lethality. Pharmacological suppression of cell-autonomous interferon responses or transcriptional silencing of cytosolic nucleic acid sensing PRR were also sufficient to abrogate ADAR1 dependency in BRCA1-mutant cells, in line with autocrine interferon poisoning playing a central part in this synthetic lethality. Our findings provide a preclinical rationale for assessing ADAR1-targeting agents in BRCA1/2-mutant cancers, and introduces a conceptually novel approach to synthetic lethal treatments, which exploits tumor cell-intrinsic cytosolic immunity as a targetable vulnerability of cancer cells.

New Fig. 4M. Graphical model summarizing our observations. PRR, pattern recognition receptors.

To further explain these observations, we now also provide evidence that *BRCA1*-mutant cells have a constitutively elevated interferon-stimulated gene (ISG) signature and enhanced interferon-driven PRR expression (**New Suppl. Fig. 8H, I** below; LFC of 1.66 for *BRCA1*-Mut vs. *BRCA1*-Rev; $P < 0.0001$) – a phenotype which has also been observed in *BRCA1*-mutant tumors (Parkes JNCI 2016; Bruand Cell Rep 2021; Reisländer Nat Commun 2019). As is the case with ISG signature-high tumors (Liu Nat Med 2019), *BRCA1*-mutant cancer cells therefore elicit stronger cytosolic nucleic acid sensing responses also due to enhanced PRR expression driven by chronic interferon signaling.

We have now incorporated this new data in the manuscript, and have completed the Results section accordingly:

Results section, page 13:

Secondly, to further explore the selective activation of an interferon response in BRCA1-mutant cells, we compared the ISG signature of SUM149 BRCA1-Mut and BRCA1-Rev cells following ADAR1 silencing using RNA-Seq. This revealed (i) a constitutively elevated ISG signature in SUM149 BRCA1-Mut cells in absence of ADAR1 silencing (Supplementary Fig. 8H; LFC = 1.66 for BRCA1-Mut vs. BRCA1-Rev, $P < 0.0001$), and (ii) a selective upregulation of the ISG signature upon ADAR1 silencing in SUM149 BRCA1-Mut cells (Supplementary Fig. 8I; LFC = 0.34 for siADAR1 vs. siCTRL in BRCA1-Mut, $P = 0.0029$; LFC = 0.16 for siADAR1 vs. siCTRL in BRCA1-Rev, $P = 0.0617$, not significant), thereby confirming our previous observations (Fig. 4A-F).

New Suppl. Fig. 8H. Heatmap showing transcript-level changes of genes within the interferon-stimulated genes (ISG) signature in SUM149 *BRCA1*-Mut and *BRCA1*-Rev cells. Color mapping indicates z-scores of gene expression based on RNA-Seq. $n=3$. **I.** Heatmap showing transcript-level changes of genes within the interferon-stimulated genes (ISG) signature in SUM149 *BRCA1*-Mut and *BRCA1*-Rev cells transfected with *ADAR1* siRNA. Color mapping indicates z-scores of gene expression based on RNA-Seq. $n=3$.

Specific comments:

1. As discussed above, the data in Fig. 4A-4F are surprising. Why are *BRCA1* mutant cells are more responsive to dsRNA? The observations in these panels are not explained.

Our response: Please see our response above to the previous same point.

2. In Fig. 4G and H, it is surprising that depleting *RIG1*, *MDA5*, *LGP2*, and *PKR* individually reversed the *ADAR1*-*BRCA1* synthetic lethality similarly. Should these proteins respond to dsRNA through independent pathways? If so, one would expect some redundancy among these proteins. It is surprising that depletion of all these proteins individually can suppress the synthetic lethality substantially.

Our response: Our data do show that depleting *RIG1*, *MDA5*, *LGP2* and *PKR* individually reverses the synthetic lethality. Still, this reversal is only partial (Fig. 4G, H; Suppl. Fig. 8J-M), as opposed to the complete abrogation observed with *JAK/STAT* pathway inhibitors (Fig. 4J, K; Suppl. Fig. 9A-D), which act downstream of the above PRR. In line with the reviewer's comment, such partial reversal suggests some degree of redundancy and/or crosstalk (Cao Nat Rev Immunol 2016) between these PRR sensing pathways.

To best explore this, we have assessed the expression level of these PRR upon silencing of each other PRR (**New Fig. 4I**). We found that silencing *cGAS* reduced *RIG1*, *MDA5*, and *LGP2*

expression levels; conversely, silencing of either RIG1, MDA5, or LGP2 reduced the expression levels of all three RNA sensors (as well as phosphorylation levels of PKR), but not those of cGAS (**New Fig. 4I**). Together with the complete reversal of the SL observed upon exposure to JAK/STAT pathway inhibitors, this overall supports some level of redundancy among these proteins, as suggested by the reviewer.

We have now included this new data and modified the manuscript accordingly to discuss these aspects:

New Fig. 4I. Western blot of SUM149 *BRCA1*-Mut and *BRCA1*-Rev cells subjected to co-transfection with *ADAR1* siRNA and one of a series of siRNAs targeting RNA and RNA:DNA hybrid sensors. $n=3$.

Results section, page 14:

Of note, silencing of cGAS also reduced the expression levels of RIG1, MDA5, and LGP2 while conversely, silencing of either RIG1, MDA5, or LGP2 reduced the expression levels of all three dsRNA sensors (as well as phosphorylation levels of PKR) but not those of cGAS (Fig. 4I). This indicated an interdependence between cytosolic dsRNA, RNA:DNA and DNA sensing pathways as previously described⁵¹, and suggested some level of redundancy among these proteins in mediating the BRCA1/2–ADAR1 SL effects.

3. In Fig. 3M and N, RNaseH1 rescued the viability of siADAR1-BRCA1 mutant cells almost completely. From the extents of rescue, the contribution of R-loops to synthetic lethality is similar to those of dsRNA sensors. This is not consistent with the conclusion that dsRNA-induced autocrine interferon responses are the primary cause of synthetic lethality.

Our response: We agree with the reviewer that our data support a key contribution of R-loops to the *BRCA1/2–ADAR1* SL, and have now revised our Abstract to better reflect this:

Abstract, page 2:

ADAR1 is an RNA editing enzyme which prevents autoimmunity by blocking interferon responses triggered by cytosolic RNA sensors, and is a potential target in immuno-oncology. However, predictive biomarkers for ADAR1 inhibition are lacking. Using multiple in vitro and in vivo systems, we show that BRCA1/2 and ADAR1 are synthetic lethal, and that ADAR1 activity is upregulated in BRCA1/2-mutant cancers. ADAR1 depletion in BRCA1-mutant cells caused an increase in R-loops and consequently, an

upregulation of cytosolic nucleic acid sensing pattern recognition receptors (PRR), events which were associated with a tumor cell-autonomous type I interferon and integrated stress response. This ultimately caused autocrine interferon poisoning. Consistent with the key role of R-loops in this process, exogenous RNase H1 expression reversed the synthetic lethality. Pharmacological suppression of cell-autonomous interferon responses or transcriptional silencing of cytosolic nucleic acid sensing PRR were also sufficient to abrogate ADAR1 dependency in BRCA1-mutant cells, in line with autocrine interferon poisoning playing a central part in this synthetic lethality. Our findings provide a preclinical rationale for assessing ADAR1-targeting agents in BRCA1/2-mutant cancers, and introduces a conceptually novel approach to synthetic lethal treatments, which exploits tumor cell-intrinsic cytosolic immunity as a targetable vulnerability of cancer cells.

We think the activation of cytosolic nucleic acid sensing PRR and the subsequent interferon response are also key to the SL, but operate downstream of the increase in R-loops. Our data that show that the SL is reversed by the silencing of various PRR (Fig. 4G-I; Suppl. Fig. 8J-M) or exposure to JAK/STAT pathway inhibitors (Fig. 4J, K; Suppl. Fig. 8N; Suppl. Fig. 9A-D; Suppl. Fig. 10) does not necessarily mean that the PRR aspect of the SL operates independently of an increase in R-loops. In line with this idea, there is prior evidence to suggest that increased R-loops activate nucleic acid sensing PRR, including:

- (i) Sensing of R-loop-derived cytosolic RNA:DNA hybrids by cGAS (Crossley Nature 2023; Mankan EMBO J 2014), RIG1, MDA5 (Crossley Nature 2023), PKR (Wang Genome Res 2018), TLR3 (Crossley Nature 2023), TLR9 (Rigby EMBO J 2014) or NLRP3 (Vanaja PNAS 2014), which can individually or collectively cross-activate other PRR pathways (Cao Nat Rev Immunol 2016);
- (ii) Sensing of R-loop-derived cytosolic DNA by cGAS (Schumann J Exp Med 2023; Mackenzie EMBO J 2016; Chatzidoukaki Sci Adv 2021; Maxwell Cell 2024; Zannini Cell Mol Life Sci 2024);
- (iii) Sensing of R-loop-derived cytosolic RNA by MDA5 (Schumann J Exp Med 2023);

We have now amended our mechanistic model (**New Fig. 4M**) to better highlight this sequential contribution of R-loops and cytosolic nucleic acid sensing PRR to the SL:

New Fig. 4M. Graphical model summarizing our observations. PRR, pattern recognition receptors.

In light of all of the above, we have also modified the manuscript as follows:

Results section, page 11:

RH1 overexpression in BRCA1-mutant cells reversed their sensitivity to ADAR1 silencing in colony-formation assay (Fig. 3M, N) and to a lesser extent in short-term survival assays (Supplementary Fig. 6A-E).

Results section, page 16:

Since both RH1 overexpression and PRR knockdown reversed the BRCA1/2–ADAR1 SL, we assessed the contribution of R-loops to PRR-driven innate immune responses. To do this, we replicated western blots of dsRNA sensors, type I interferon and ISR markers in SUM149 BRCA1-isogenic cells subjected to (i) ADAR1 silencing and/or (ii) RH1 overexpression and/or (iii) JSPi exposure. Whilst BRCA1-mutant cells were unable to activate a type I interferon response upon ADAR1 silencing when exposed to JSPi (Supplementary Fig. 10A) or co-silencing of RIG1 or LGP2 (Supplementary Fig. 10B), RH1 overexpression also counteracted (albeit partially) the upregulation of PRR, type I interferon and integrated stress response markers elicited upon ADAR1 silencing in BRCA1-mutant cells (Supplementary Fig. 10A, B). Consistent with this, short-term survival assays conducted in the same conditions revealed that RH1 overexpression and exposure to JSPi both contributed to reverse the cytotoxic effects of ADAR1 siRNA in SUM149 BRCA1-Mut cells (Supplementary Fig. 10C).

Altogether, this data supported the idea that increased R-loop burden elicited by ADAR1 suppression associates with PRR-driven cell-autonomous interferon poisoning in BRCAm cells, thereby ultimately causing the BRCA1/2–ADAR1 SL.

Discussion section, page 17:

Mechanistically, we show that the BRCA1/2–ADAR1 SL relies upon R-loop-associated DNA damage (Fig. 3) and the selective activation of an R-loop- and PRR-driven cell-autonomous innate immune response in BRCAm cells (Fig. 4), providing the first evidence for autocrine interferon poisoning as a mechanism of BRCA1/2 SL⁴⁶.

Importantly, these findings delineate a distinctive SL mechanism, which is unlike those of other BRCA1/2-associated SL, including the BRCA1/2–PARPi SL. In line with the recently described immunogenic potential of R-loops^{55–62}, our results support a model (Fig. 4M) in which unresolved R-loops elicited by the loss of ADAR1 function^{4,6,48} activate cytosolic nucleic acid sensing PRR, resulting in a toxic cell-autonomous innate immune response which generates the SL phenotype. ADAR1p150 plays an essential role in this SL and could be responsible for the suppression of nuclear R-loops in BRCAm cells, consistent with its nucleocytoplasmic distribution^{63,64}.

4. The data in Fig. S10A is not consistent with Fig. 3N. The rescue of cell survival by RNaseH1 expression was nearly 100% in Fig. 3N, but only ~50% in Fig. S10A. The partial rescue in S10A by RNaseH1 is the reason why JAK1/2i can exert additional rescuing effects.

Our response: This is a fair observation and we apologize for not having appropriately explained this in our primary submission. The difference initially observed between original Fig. 3N and Fig. S10A relates to the timing of the readout (long-term vs short-term, respectively). We could unfortunately not compare the levels of rescue induced by RNase H1 overexpression vs. JAK-STAT pathway inhibitors exposure in long-term assays, because of the intrinsic cytotoxic effects of the latter. Therefore, we agree with the reviewer and cannot exclude that JAK1/2 inhibitors exert additional rescuing effects at early timepoints.

We have now highlighted these differences in the manuscript text to better explain our observations:

Results section, page 11:

RH1 overexpression in BRCA1-mutant cells reversed their sensitivity to ADAR1 silencing in colony-formation assay (Fig. 3M, N) and to a lesser extent in short-term survival assays (Supplementary Fig. 6A-E).

5. The data in Fig. S10B-D only show that the dsRNA responding pathways are still functional in the presence of RNaseH1. However, they don't necessarily mean that these dsRNA pathways are the primary cause of synthetic lethality.

Our response: We agree with the reviewer and have now modified the manuscript to correct our interpretation:

Results section, page 16:

Whilst BRCA1-mutant cells were unable to activate a type I interferon response upon ADAR1 silencing when exposed to JSPi (Supplementary Fig. 10A) or co-silencing of RIG1 or LGP2 (Supplementary Fig. 10B), RH1 overexpression also counteracted (albeit partially) the upregulation of PRR, type I interferon and integrated stress response markers elicited upon ADAR1 silencing in BRCA1-mutant cells (Supplementary Fig. 10A, B). Consistent with this, short-term survival assays conducted in the same conditions revealed that RH1 overexpression and exposure to JSPi both contributed to reverse the cytotoxic effects of ADAR1 siRNA in SUM149 BRCA1-Mut cells (Supplementary Fig. 10C).

Altogether, this data supported the idea that increased R-loop burden elicited by ADAR1 suppression associates with PRR-driven cell-autonomous interferon poisoning in BRCAm cells, thereby ultimately causing the BRCA1/2-ADAR1 SL.

End of Referee #2 comments

Reviewer #3 (Remarks to the Author): with expertise in ADAR1, RNA editing

The manuscript titled “Autocrine Interferon Poisoning Mediates ADAR1-Dependent Synthetic Lethality in BRCA1/2-Mutant Cancers” by Chabanon et al. presents novel and important findings. The study demonstrates that ADAR1, an RNA editing enzyme, and BRCA1/2 exhibit synthetic lethality. The authors explore the mechanisms underlying this relationship and find that the loss of ADAR1 causes DNA damage due to defective R-loop resolution. However, the primary mechanism involves autocrine interferon poisoning, triggered by the activation of RNA sensors (LGP2 and PKR) in BRCA1/2-mutant tumor cells. This paper has significant implications for cancer therapy, as it highlights interferon signaling as a key vulnerability and provides a preclinical rationale for targeting ADAR1 as a therapeutic strategy in BRCA1/2-mutant cancers. The authors utilize an impressive variety of systems and animal models to demonstrate their findings. I have only a few minor comments:

1. Previous studies have shown that RNA editing is indeed elevated in BRCA cancers, and this finding can complement the narrative of this paper (see Fumagalli et al. and Paz-Yaacov et al., both from 2015).

Our response: We thank the reviewer for this very relevant comment and apologize for not having initially cited these papers. We now refer to these anterior findings in the manuscript text, which reads:

Results section, page 9:

Previous studies have shown that ADAR1-mediated RNA editing was elevated in breast cancer as compared with matched, normal breast tissue ^{41,42} and compared with other types of cancers ⁴¹, but the impact of BRCA1/2 mutations on the magnitude of A-to-I RNA editing in cancer remains unknown.

2. It is well established that specific coding sites, such as AZIN, which is a target of ADAR1P150, are critical for cancer progression under certain conditions (see for example PMID: 23291631). The authors should discuss the possibility that, in addition to double-stranded RNA, a lack of editing at this target may contribute to the presented phenotype.

Our response: We thank the reviewer for this note and interesting perspective. We now discuss this possibility in the manuscript discussion, which reads:

Discussion section, page 19:

Thirdly, complementary mechanisms might also contribute to this SL. These include the antagonism of ADAR1-mediated recoding activity at specific editing sites ⁸⁰, mitotic failure due to defective ADAR1p150 function in chromosome segregation ⁸¹, or ZBP1-dependent necroptosis following defective editing of Alu duplex RNA ⁸²⁻⁸⁴. Yet, the complete abrogation of SL effects upon JAK/STAT pathway inhibition suggests a central role of interferon-dependent, “viral mimicry” responses in mediating the BRCA1/2–ADAR1 SL phenotype. Whether these alternative mechanisms may occur synergistically or in a context-specific fashion would deserve further exploration.

3. In Figure #2, the authors present the editing levels. To demonstrate the background levels as well, it would be beneficial to include the editing levels of other mismatches (e.g., C-to-T).

Our response: The editing levels of all other edition types have been computed and the corresponding graphs are now supplied in the figures (see **New Suppl. Fig. 4G-P** and Fig. 2F, G below).

New Suppl. Fig. 4G-P. RNA editing levels of all possible types of editions displayed as RNA editing index (G-K) or number of RNA editing sites (L-P) in SUM149 *BRCA1*-Mut and *BRCA1*-Rev cells transfected with *ADAR1* siRNA. Cells were transfected with control, non-targeting siRNA (siCTRL), or with either *ADAR1* siRNA SMARTpool (P) or an individual *ADAR1* siRNAs (#1). 60 hours after transfection, RNA extracts were generated and subjected to whole-transcriptome sequencing. Bar plots indicate mean \pm SD; $n=3$, two-way ANOVA post hoc Tukey's test.

Fig. 4F, G. A-to-I RNA editing levels displayed as RNA editing index (F) or number of RNA editing sites (G) in SUM149 *BRCA1*-Mut and *BRCA1*-Rev cells transfected with *ADAR1* siRNA. Bar plots indicate mean \pm SD; $n=3$, two-way ANOVA post hoc Tukey's test.

The manuscript has been amended accordingly to present A-to-I RNA editing levels in comparison with editing levels of other mismatches:

Results section, pages 9-10:

As expected, silencing of *ADAR1* caused a dramatic reduction of RNA editing in both *BRCA1*-mutant and -revertant cells as measured by a decreasing A-to-I RNA Editing Index (REI; **Fig. 2F**) and fewer A-

to-I RNA Editing Sites (RES; Fig. 2G). More importantly, these analyses revealed a significantly higher A-to-I RNA editing activity of BRCA1-mutant cells compared with BRCA1-revertant cells (mean number of 143,169 vs. 62,630 RES, respectively; $P < 0.0001$, two-way ANOVA; Fig. 2F, G), despite similar ADAR1 protein levels (Supplementary Fig. 1B, D, 4F). This difference was further specific to A-to-I RNA editing, as levels of editing observed for all other types of editing remained extremely low and unchanged (mean number of RES $< 1,200$ for C-to-T transitions and close to null for all transversions; Supplementary Fig. 4G-P).

End of Referee #3 comments

REVIEWER COMMENTS

Reviewer #1 (Remarks to the Author): with expertise in BRCA-mutant cancer

The authors have satisfactorily addressed all the key concerns. I look forward to seeing this exciting work in-print.

Our response: We thank the reviewer for taking the time to review our work.

End of Referee #1 comments

Reviewer #2 (Remarks to the Author): with expertise in BRCA-mutant cancer, synthetic lethality

The authors have done a good job addressing most of my comments.

Our response: We thank the reviewer for acknowledging the quality of our rebuttal.

However, I am still confused about how R-loops activate autocrine interferon poisoning. Although RNaseH1 reversed the synthetic lethality effectively, several of the PRRs involved are not known sensors of R-loops. Even in the revised model figure, it is not clear how R-loop burden activates the whole group of PRRs.

Our response: In the manuscript, we show that when ADAR1 is silenced in *BRCA1*-mutant cells, overexpression of RNase H1, which reduces R-loop burden, also reduces the upregulation of canonical dsRNA-sensing PRRs (e.g. RIG1, MDA5, PKR), type I interferon and integrated stress response markers (Suppl. Fig. 10). At face value, this indicates that directly or indirectly (and we suspect sometimes indirectly), it is the increase in R-loops that leads to activation of these PRR pathways. Others (described below) have made similar observations that increasing R-loops activates dsRNA-sensing PRRs, albeit not in the context of a *BRCA1/2-ADAR1* synthetic lethality. Although neither we, nor any of these other investigators, understand with precise clarity how R-loops control the activation status of dsRNA-sensing PRRs, our data establish a causative link between an increase in R-loops, downstream PRR activation and the *BRCA1/2-ADAR1* synthetic lethality.

As alluded to above, it is interesting to speculate as to whether R-loops directly or indirectly activate specific PRRs. Whilst some of the activated PRRs are known to detect RNA:DNA hybrids similar to those formed at R-loops, e.g. cGAS (Crossley Nature 2023), RIG1 or PKR (Wang Genome Res 2018), other PRRs have not yet been described to do so (e.g. LGP2, MDA5), suggesting that either the substrate specificity of PRRs might be less discrete than expected and/or activation of some PRRs, through crosstalk, results in the activation of others.

As detailed in our previous rebuttal, the idea that an increase in R-loops causes activation of canonical dsRNA-sensing PRR pathways has precedence in the literature, and is compatible with a possible co-existence of various R-loop-derived immunogenic cytosolic nucleic acids species, including: (i) cytosolic RNA:DNA hybrids, which beyond cGAS (Crossley Nature 2023; Mankan EMBO J 2014) can also be bound by RIG1 and PKR (Wang Genome Res 2018) and elicit innate immune responses mediated by RIG1, MDA5, TLR3 (Crossley Nature 2023), TLR9 (Rigby EMBO J 2014) or NLRP3 (Vanaja PNAS 2014); (ii) R-loop-derived cytosolic RNA, which activates MDA5 (Schumann J Exp Med 2023); and (iii) R-loop-derived cytosolic DNA, which activates cGAS (Mackenzie EMBO J 2016; Chatzidoukaki Sci Adv 2021; Maxwell Cell 2024;

Zannini Cell Mol Life Sci 2024; Schumann J Exp Med 2023) and as a consequence, can also trigger other PRR pathways through crosstalk (Cao Nat Rev Immunol 2016).

We are more than happy to further detail all of the above in an expanded Discussion of the manuscript as well as to outline how subsequent investigation could resolve what the interplay between different PRRs is that pertains to the *BRCA1/2–ADAR1* synthetic lethality.

The Western blots in the new Fig. 4I makes the interpretation more difficult. If siRNAs targeting individual PRRs actually reduced multiple PRRs simultaneously, it would be hard to pinpoint which PRR is actually required.

Our response: We agree with the reviewer that the intrinsic interdependence of PRR pathways makes it difficult to identify which PRR is specifically involved in the *BRCA1/2–ADAR1* synthetic lethality. Still, the extent to which synthetic lethal effects are reversed upon PRR silencing varies across PRRs and is most robust upon silencing of LGP2, PKR or cGAS (Fig. 4G, H; Suppl. Fig. 8K-M), suggesting that these PRRs are more important for the synthetic lethality as compared with others.

We have previously highlighted this in the revised manuscript text:

Results section, pages 13-14:

*We next directly assessed whether the *BRCA1/2–ADAR1* SL might depend upon cytosolic nucleic acid sensing, and if so, sought to identify which PRR might be involved. To this aim, we conducted co-silencing experiments in which SUM149 *BRCA1*-isogenics were transfected with either *ADAR1* siRNA and a non-targeting control siRNA or with *ADAR1* siRNA and one of a series of siRNAs targeting cytosolic dsRNA sensors (*RIG1*, *MDA5*, *LGP2*, *PKR*) or the non-canonical RNA:DNA hybrid sensor *cGAS*. In these assays, we found that the concomitant silencing of *ADAR1* with either *RIG1*, *MDA5*, *LGP2*, *PKR* or *cGAS* resulted in a significant reversal of *ADAR1* SL effects in SUM149 *BRCA1*-Mut cells (Fig. 4G, H), while the knockdown of each sensor individually had little effect on cell viability in absence of *ADAR1* silencing (Supplementary Fig. 8J-M). If the extent of this rescue was modest with *MDA5* and *RIG1* knockdown, *LGP2*, *PKR* or *cGAS* knockdown elicited a substantial reversal of the SL, akin to the effect of type I interferon receptor (*IFNAR1*) silencing, used as a positive control (Fig. 4G, H).*

*Also, if the *PKR* mediated effects are only driven by dsRNA, the complete reversal of SL by *RNaseH1* is surprising (Fig. 3N).*

Our response: Similar to the above discussion, although the canonical function of *PKR* is sensing of dsRNA, there is prior empirical evidence showing that *PKR* binds RNA:DNA hybrids (Wang Genome Res 2018), suggesting that this canonical view does not explain the entirety of *PKR* biology.

*The *RNaseH1* rescue data in Fig. 3N and S10C are not very consistent. The reversal of SL by *RNaseH1* becomes partial in Fig. S10C.*

Our response: In our prior rebuttal (which was point 4 in the reviewer's first set of comments) and revised manuscript, we addressed this issue as follows:

This is a fair observation, and we apologize for not having appropriately explained this in our primary submission. The difference initially observed between original Fig. 3N and Fig. S10C relates to the timing of the readout (long-term vs short-term, respectively). We could unfortunately not compare the levels of rescue induced by *RNase H1* overexpression vs. *JAK-STAT* pathway inhibitors exposure in long-term assays, because of the intrinsic cytotoxic effects of the latter. Therefore, we agree with the reviewer and cannot exclude that *JAK1/2* inhibitors exert additional rescuing effects at early timepoints.

We have also previously highlighted these differences in the manuscript text to better explain our observations:

Results section, page 11:

RH1 overexpression in BRCA1-mutant cells reversed their sensitivity to ADAR1 silencing in colony-formation assay (Fig. 3M, N) and to a lesser extent in short-term survival assays (Supplementary Fig. 6A-E).

Overall, this is a very nice study, but some gaps remain noticeable.

Our response: We thank the reviewer for this very positive comment, and acknowledge that our study does not fully decipher the links between R-loops and activation of nucleic acid sensing PRRs in the context of the *BRCA1/2-ADAR1* synthetic lethality – although this has been studied by others in different contexts. We have acknowledged these limitations in the revised manuscript discussion:

Discussion section, pages 17-19:

In line with the recently described immunogenic potential of R-loops⁵⁵⁻⁶², our results support a model (Fig. 4M) in which unresolved R-loops elicited by the loss of ADAR1 function^{4,6,48} activate cytosolic nucleic acid sensing PRR, resulting in a toxic cell-autonomous innate immune response which generates the SL phenotype. ADAR1p150 plays an essential role in this SL and could be responsible for the suppression of nuclear R-loops in BRCAm cells, consistent with its nucleocytoplasmic distribution^{63,64}. Still, some elements of the BRCA1/2-ADAR1 SL mechanism remain to be explored.

First, the PRR specificity of the BRCA1/2-ADAR1 SL remains unclear. Our results indicate that LGP2, PKR and cGAS silencing most reproducibly rescue ADAR1 SL effects, suggesting a contribution of multiple cytosolic nucleic sensing pathways. However, considering the interdependence and possible redundancy of these pathways, other PRR might also contribute to these SL effects (e.g. TLR3 or ZBP1^{55,65}). Intriguingly, silencing of DHX9 (a PRR which both suppresses cytosolic dsRNA sensing^{59,66} and bears a nuclear RNA helicase activity against R-loops⁶⁷) did not elicit BRCA1-dependent SL effects in our original screen, suggesting a possible redundancy of DHX9 function (i) at the nuclear level, with that of other helicases for the resolution of R-loops and/or (ii) at the cytosolic level, with that of other PRR for the suppression of dsRNA sensing-mediated responses.

Related to this, the exact nature of cytosolic nucleic acid species triggering cell-autonomous interferon poisoning remains to be defined. Although our data point to a central role of R-loops in the mechanism of this SL, multiple cytosolic nucleic sensing PRR – not restricted to those described as putative RNA:DNA hybrid sensors (i.e. cGAS^{55,68}, TLR3⁵⁵, TLR9⁶⁹ and NLRP3⁷⁰), are activated by ADAR1 silencing in BRCA1/2-mutant cells. Prior work indicates that R-loop-derived cytosolic nucleic acids – including DNA, dsRNA, and RNA:DNA hybrids, can trigger interferon responses via DNA- and dsRNA-sensing PRR⁵⁵⁻⁶², and conversely, that canonical cytosolic DNA and dsRNA sensors can detect RNA:DNA hybrids^{55,71}, suggesting a possible contribution of several cytosolic nucleic acid species to the SL. Besides, others have shown that cytosolic dsRNA arises from R-loop-prone genomic regions, including telomeres (via telomeric repeat-containing RNA, TERRA)^{4,65,72,73}, and micronuclei (via aberrant transcription of chromosomes trapped in micronuclei)^{48,74}. These latter findings could be consistent with our observations that an increased R-loop burden also, in the context of the BRCA1/2-ADAR1 SL, activates dsRNA-sensing PRR. Interestingly, our observation that RNase H1 overexpression does not fully abrogate the type I interferon response elicited by ADAR1 silencing in BRCA1-mutant cells is in line with the notion that cytosolic RNA:DNA hybrids originate from a subset of long-lived nuclear R-loops that are partially RNase H-resistant⁵⁵.

Secondly, the characterization of R-loop as the trigger of ADAR1-dependent SL effects would require further investigation to determine the nature (promoter-paused vs. elongation-associated R-loops⁷⁵), genomic location (promoter proximal, exonic or non-coding regions) and functional context (co-transcriptional vs. DNA repair-associated R-loops at DSBs⁷⁶⁻⁷⁸ or replication forks⁷⁹) of R-loops involved in the BRCA1/2-ADAR1 SL. The use of orthogonal methods to visualize (e.g. catalytically-

inactive mutant RNase H1 protein) and capture both native (e.g. R-loop CUT&Tag) and ex cellulo-isolated R-loops (e.g. DRIP-Seq) would allow a more comprehensive profiling of the R-loop landscape in response to impaired ADAR1 function.

Thirdly, complementary mechanisms might also contribute to this SL. These include the antagonism of ADAR1-mediated recoding activity at specific editing sites⁸⁰, mitotic failure due to defective ADAR1p150 function in chromosome segregation⁸¹, or ZBP1-dependent necroptosis following defective editing of Alu duplex RNA⁸²⁻⁸⁴. Yet, the complete abrogation of SL effects upon JAK/STAT pathway inhibition suggests a central role of interferon-dependent, “viral mimicry” responses in mediating the BRCA1/2–ADAR1 SL phenotype. Whether these alternative mechanisms may occur synergistically or in a context-specific fashion would deserve further exploration.

End of Referee #2 comments

Reviewer #3 (Remarks to the Author): with expertise in ADAR1, RNA editing

The authors have addressed all of my concerns and have undertaken much of the necessary work.

Our response: We thank the reviewer for taking the time to review our work.

End of Referee #3 comments